**The HTAP_v3 emission mosaic: merging regional and global monthly emissions (2000-2018) to support air quality modelling and policies**

Monica Crippa[1], Diego Guizzardi[1], Tim Butler[2], Terry Keating[3], Rosa Wu[16], Jacek Kaminski[4], Jeroen Kuenen[5], Junichi Kurokawa[6], Satoru Chatani[15], Tazuko Morikawa[7], George Pouliot[17], Jacinthe Racine[8], Michael D. Moran[16], Zbigniew Klimont[9], Patrick M. Manseau[8], Rabab Mashayekhi[8], Barron H. Henderson[17], Steven J. Smith[10], Harrison Suchyta[10], Marilena Muntean[1], Efisio Solazzo[11], Manjola Banja[1], Edwin Schaaf[1], Federico Pagani[12], Jung-Hun Woo[13,14], Jinseok Kim[14], Fabio Monforti-Ferrario[1], Enrico Pisoni[1], Junhua Zhang[16], David Niemi[8], Mourad Sassi[8], Tabish Ansari[2], Kristen Foley[17]

[1]European Commission, Joint Research Centre (JRC), Ispra, Italy

[2]Research Institute Sustainability - Helmholtz Centre Potsdam (RIFS Potsdam), Potsdam, 14467, Germany

[3]U.S. Environmental Protection Agency, Washington DC 20460, USA

[4]Institute of Environmental Protection, National Research Institute, Poland

[5]Department of Climate, Air and Sustainability, TNO, Utrecht, The Netherlands

[6]Asia Center for Air Pollution Research (ACAP), 1182 Sowa, Nishi-ku, Niigata-shi 950-2144, Japan

[7]Japan Automobile Research Institute (JARI)

[8]Air Quality Policy-Issue Response Section, Canadian Centre for Meteorological and Environmental Prediction, Environment and Climate Change Canada (ECCC), Dorval, Quebec, Canada

[9]International Institute for Applied Systems Analysis (IIASA), Laxenburg, Austria

[10]Joint Global Change Research Institute, Pacific Northwest National Lab, College Park, MD, USA

[11]UniSystems Company, Milan (Italy)

[12]GFT Italia S.r.l., VIA SILE, 20139 Milano, Italy

[13]Department of Civil and Environmental Engineering, Konuk University, Seoul, South Korea

[14]Department of Technology Fusion Engineering, Konuk University, Seoul, South Korea

[15]National Institute for Environmental Studies (NIES), Tsukuba, 305-8506, Japan

[16]Air Quality Research Division, Environment and Climate Change Canada, Toronto, Ontario, Canada

[17]U.S. Environmental Protection Agency, North Carolina, USA

Correspondence to: Marilena.muntean@ec.europa.eu, monica.crippa@ext.ec.europa.eu

**Abstract.** This study, performed under the umbrella of the Task Force on Hemispheric Transport of Air Pollution (TF-HTAP), responds to the need of the global and regional atmospheric modelling community of having a mosaic emission inventory of air pollutants that conforms to specific requirements: global coverage, long time series, spatially distributed

emissions with high time resolution, and a high sectoral resolution. The mosaic approach of integrating official regional emission inventories based on locally reported data, with a global inventory based on a globally consistent methodology, allows modellers to perform simulations of a high scientific quality while also ensuring that the results remain relevant to policymakers.

HTAP_v3, an ad-hoc global mosaic of anthropogenic inventories, has been developed by integrating official inventories over specific areas (North America, Europe, Asia including Japan and Korea) with the independent Emissions Database for Global Atmospheric Research (EDGAR) inventory for the remaining world regions. The results are spatially and temporally distributed emissions of $SO_2$, NOx, CO, NMVOC, $NH_3$, $PM_{10}$, $PM_{2.5}$, Black Carbon (BC), and Organic Carbon (OC), with a spatial resolution of 0.1 x 0.1 degree and time intervals of months and years covering the period 2000-2018 (https://doi.org/10.5281/zenodo.7516361, https://edgar.jrc.ec.europa.eu/dataset_htap_v3). The emissions are further disaggregated to 16 anthropogenic emitting sectors. This paper describes the methodology applied to develop such an emission mosaic, reports on source allocation, differences among existing inventories, and best practices for the mosaic compilation. One of the key strengths of the HTAP_v3 emission mosaic is its temporal coverage, enabling the analysis of emission trends over the past two decades. The development of a global emission mosaic over such long time series represents a unique product for global air quality modelling and for better-informed policy making, reflecting the community effort expended by the TF-HTAP to disentangle the complexity of transboundary transport of air pollution.

## 1 Introduction

Common international efforts have procured an agreement to reduce global air pollutant emissions. For this purpose, the United Nations Economic Commission for Europe (UNECE) Convention on Long Range Transboundary Air Pollution (CLRTAP) and the Task Force on Hemispheric Transport of Air Pollution (TF-HTAP) have been instrumental in developing the understanding of intercontinental transport of air pollution  and thus contributing to the reduction of key pollutants in Europe and North America.

The success of CLRTAP is based on meeting strict reduction targets for pollutant releases. Therefore, evaluating the resulting implications of these reductions requires an ongoing improvement of global emission inventories in terms of emission updating and of methodological refinements. These aspects are instrumental to gain understanding of transboundary air pollution processes and drivers and to measure the effectiveness of emissions reduction and air quality mitigation policies. New guidance is available to achieve further emission reductions across all emitting sectors. For example, the 2019 establishment of the Task Force for International Cooperation on Air Pollution, which is intended to promote international collaboration for preventing and reducing air pollution and improving air quality globally (UNECE, 2021). As part of the ongoing effort by CLRTAP to reduce emissions and to set out more effective and accountable mitigation measures, the 2005 Gothenburg Protocol (UNECE, 2012) has been revised, including the review of the obligations in relation to emission reductions and mitigation measures (e.g., black carbon and ammonia) and the review of the progress towards achieving the environmental and health objectives of the Protocol.

The Task Force on Hemispheric Transport of Air Pollution (TF-HTAP) of the Convention has a mandate to promote the scientific understanding of the intercontinental transport of air pollution to and from the UNECE area (https://unece.org/geographical-scope), to quantify its impacts on human health, vegetation and climate, and to identify emission mitigation options that will shape future global policies.

This paper describes and discusses a consistent global emission inventory of air pollutants emitted by anthropogenic activities. This important database has been developed to assess the contribution of anthropogenic air pollution emission sources within and outside the UNECE-area through atmospheric modelling. This inventory has been compiled based on officially reported emissions, and an independent global inventory where officially reported emissions are not used. This harmonised emissions "mosaic" dataset, hereafter referred to as the HTAP_v3, contains annual and monthly:

- emission time series (from 2000 to 2018) of $SO_2$, NOx (expressed as $NO_2$ mass unit), CO, NMVOC, $NH_3$, $PM_{10}$, $PM_{2.5}$, BC, OC by emitting sector and country, and
- spatially distributed emissions on a global grid with spatial spacing of 0.1x0.1 degree.

The creation of a global emission mosaic requires the harmonisation of several data sources, detailed analysis of contributing sectors for the different input inventories, development of data quality control procedures, and a robust and consistent gap-filling methodology when lacking information. The development of HTAP_v3 builds upon the previous experience of the HTAPv1 (Janssens-Maenhout et al., 2012) and HTAPv2.2 (Janssens-Maenhout et al., 2015) global inventories. HTAP_v3, as requested by the TF-HTAP modelling community, provides a more refined sectoral disaggregation compared to the previous HTAP emission mosaics. It also includes tools (https://edgar.jrc.ec.europa.eu/htap_tool/) that allow the extraction of emission data over selected domains (detailed later in section 4).

This paper describes the development the HTAP_v3 database as a global anthropogenic air pollutant emissions inventory mosaic for the period 2000-2018. The HTAP_v3 mosaic has been composed by integrating official, spatially distributed emissions data from CAMS-REG-v5.1 (Kuenen et al., 2022), US EPA (U.S. Environmental Protection Agency, 2021b, a), Environment and Climate Change Canada (ECCC) (NPRI, 2017), REAS, CAPSS-KU, and JAPAN (https://www.env.go.jp/air/osen/pm/info.html) (Kurokawa and Ohara, 2020; Chatani et al., 2018; Chatani et al., 2020) inventories. As the information gathered from the official reporting covers only part of the globe, HTAP_v3 has been completed using emissions from the Emissions Database for Global Atmospheric Research (EDGAR) version 6.1 (https://edgar.jrc.ec.europa.eu/dataset_ap61).

One of the key strengths of the HTAP_v3 emission mosaic is the temporal coverage of the emissions, spanning the 2000-2018 period, enabling the analysis of emission trends over the past two decades. The development of a global emission mosaic over such long time series represents a unique product for air quality modelling and for better-informed policy making, reflecting the effort of the TF-HTAP community to improve understanding of the transboundary transport of air pollution. The year 2000 was chosen as the start year since it often represents the year from which complete datasets of annual air pollutant emissions can be generated. It also represents a turning point for several emerging economies (e.g., China) and the strengthening of mitigation measures in historically developed regions (e.g., EU, USA, etc.).

The two previous HTAP emission mosaics had limited temporal coverage. HTAPv1 covered the period 2000-20005 with annual resolution (https://edgar.jrc.ec.europa.eu/dataset_htap_v1, (Janssens-Maenhout et al., 2012)), while HTAPv2.2 covered two recent years (2008 and 2010), but with monthly resolution (Janssens-Maenhout et al., 2015) (https://edgar.jrc.ec.europa.eu/dataset_htap_v2). However, the needs of the TF-HTAP modelling community are continuously evolving to both foster forward-looking air quality science and produce more fit-for-purpose analyses in support of efficient policy making. HTAP_v3 therefore not only covers the time period of the previous HTAP phases, but also

extends it forward by almost a decade, to provide the most up-to-date picture of global air pollutant emission trends. Another distinguishing feature of the HTAPv3 mosaic is a considerably higher sectoral resolution than previous iterations of the HTAP mosaic inventories (section 2.2), enabling more policy-relevant use of the inventory.

The methodology and data sources for the HTAP_v3 emission mosaic are described in section 2. The long-time coverage of two decades, allows comprehensive trend analysis (see section 3), the HTAP_v3 data format and data-set access are presented in section 4 and conclusions are provided in section 5.

## 2 HTAP_v3 emission mosaic overview: data sources, coverage, and methodology

### 2.1 Data input

The HTAP_v3 mosaic is a database of monthly- and sector-specific global air pollutant emission gridmaps developed by integrating spatially explicit regional information from recent officially-reported national or regional emission inventories. Data from six main regional inventories were integrated into HTAP_v3, which covered only North America, Europe, and a portion of Asia (including Japan, China, India, and South Korea) (Fig.1). The geographical domain covered by each of these inventories is depicted in Fig. 1, while further details on each contributing inventory are presented in section 2.3. The emissions for all other countries, international shipping and aviation (international and domestic) have been retrieved from the Emissions Database for Global Atmospheric Research (EDGARv6.1, https://edgar.jrc.ec.europa.eu/dataset_ap61) as represented by the grey areas in Fig.1. Depending on the pollutant, more than half of global emissions are provided by region-specific inventories, while the remaining contribution is derived from the EDGAR global inventory as reported in the bar graph of Fig.1, where the share of each individual inventory to global emissions is represented. For all pollutants, the Asian domain is contributing most to global emissions, hence the importance of having accurate emission inventories for this region.

Recent literature studies (Puliafito et al., 2021; Huneeus et al., 2020; Álamos et al., 2022; Keita et al., 2021; MEIC, 2022) document additional regional/local inventories which may contribute to future updates of HTAP_v3, in particular extending the mosaic compilation to regions in the Southern Hemisphere. Considering relative hemispheric emission levels as well as the atmospheric dynamics happening in the Northern Hemisphere and regulating the transboundary transport of air pollution, the current HTAP_v3 mosaic should still satisfy the needs of the atmospheric modelling community, although improvements using latest available inventories for Africa and South America may also be considered for future updates.

Table 1 provides an overview of all data providers, in terms of geographical and temporal coverage, data format, and sectoral and pollutant data availability. Table 2 defines the HTAP_v3 sectors and corresponding IPCC codes. Table 3 further details the sector-pollutant data availability for each inventory and the gap-filling approach required for some sectors and pollutants.

### 2.2 Pollutant, spatial, temporal and sectoral coverage

The HTAP_v3 emission mosaic helps to address the transboundary role of air pollutants by providing a key input for atmospheric modellers and supporting the evaluation of environmental impact analyses for poor air quality. For this reason, HTAP_v3 provides global 0.1 x 0.1 degree emission gridmaps for all air pollutants and specifically for acidifying and

eutrophying gases (such as $SO_2$, $NH_3$, $NOx$), ozone precursors (NMVOC, CO, $NOx$), and primary particulate matter ($PM_{10}$, $PM_{2.5}$, BC, OC).

Emissions from each officially-reported inventory were submitted to HTAP on 0.1 x 0.1 degree regional gridmaps. Spatial allocation was performed to these gridmaps for each sector by each inventory group using the best available set of subsector spatial surrogate fields used by each group (e.g., https://www.cmascenter.org/sa-tools). EDGARv6.1 global gridmaps are also on a 0.1 x 0.1 degree grid.

Compared to the two previous HTAP emission mosaics, HTAP_v3 input emission gridmaps were provided with monthly time distributions to better reflect the regional seasonality of sector specific emissions (e.g., household, power generation, and agricultural activities). Information on emission peaks over certain months of the year is also a useful information for the development of territorial policies to mitigate localised emission sources in space and time (e.g., emissions from residential heating over winter months, agricultural residue burning, etc.).

The HTAP_v3 mosaic provides emissions for gaseous and particulate matter air pollutants arising from all anthropogenic emitting sectors except for wildfires and savannah burning, which represent major sources of particulate matter and CO emissions. Wildfires and savannah burning are not included in the current mosaic since community efforts are ongoing to tackle these sources specifically. Modellers can find these additional sources on several publicly available global wildfire emission datasets compiled based on the best available scientific knowledge, such as the Global Fire Emission Database (GFED, https://www.globalfiredata.org/) or the Global Wildfire Information System (GWIS, https://gwis.jrc.ec.europa.eu/). When using satellite retrieved emissions from fires, they should be treated with caution to avoid double counting the emissions released by e.g. agricultural crop residue burning activities.

HTAP_v3 provides emissions at higher sectoral disaggregation than previous HTAP experiments[1] to better understand drivers of emission trends and the effectiveness of sector-specific policy implementation. Emissions from 16 sectors are provided by the HTAP_v3 mosaic, namely: International Shipping; Domestic Shipping; Domestic Aviation; International Aviation; Energy; Industry; Fugitives; Solvent Use; Road Transport; Brake and Tyre Wear; Other Ground Transport; Residential; Waste; Agricultural Waste Burning; Livestock; and Agricultural Crops. Further details on the sector definitions as well as their correspondence with the IPCC codes (IPCC, 1996, 2006) are provided in Table 2. The selection of the number of sectors was constrained by the sectoral disaggregation of the input inventories (see Table S1). Table 3 provides the complete overview of the emission data provided by each inventory group indicating the pollutants covered for each sector and eventual gap-filling information included using the EDGARv6.1 data. Table 4 reports a summary of the main features of the 3 HTAP emission mosaics, showing the advancements achieved with this work. The high sector disaggregation available within the HTAP_v3 mosaic gives needed flexibility to modellers to include or exclude emission sub-sectors in their simulations, in particular when integrating the anthropogenic emissions provided by HTAP_v3 with other components (e.g. natural emissions, forest fires, etc.). However, we recommend particular caution when using a natural

---

[1]HTAPv1 covered 10 broad emission sectors (Aircraft, Ships, Energy, Industry Processes, Ground Transport, Residential, Solvents, Agriculture, Agriculture Waste Burning, and Waste), while even broader sectoral emissions were provided in HTAPv2.2 (Air, Ships, Energy, Industry, Transport, Residential (including waste), and Agriculture (only for $NH_3$)).

emissions model such as MEGAN (Model of Emissions of Gases and Aerosols from Nature, https://www2.acom.ucar.edu/modeling/model-emissions-gases-and-aerosols-nature-megan), which includes the estimation of NMVOC emissions from crops and soil NOx emissions (including agricultural soils) that are also provided by the HTAP_v3 mosaic.

## 2.3 Inventory overviews

In the following sub-sections, details are provided on each officially-reported inventory used to construct the HTAP_v3 emission mosaic.

### 2.3.1 CAMS-REG-v5.1 inventory

The CAMS-REG-v5.1 emission inventory was developed to support air pollutant and greenhouse gas modelling activities at the European scale. The inventory builds largely on the official reported data to the UN Framework Convention on Climate Change (UNFCCC) for greenhouse gases (for $CO_2$ and $CH_4$), and the Convention on Long-Range Transboundary Air Pollution (CLRTAP) for air pollutants. For the latter, data are collected for NOx, $SO_2$, CO, NMVOC, $NH_3$, $PM_{10}$ and $PM_{2.5}$, including all major air pollutants. For each of these pollutants, the emission data are collected at the sector level at which these are reported for the time series 2000-2018 for each year and country. The CAMS-REG inventory covers UNECE-Europe, extending eastward until 60°E, therefore including the European part of Russia. For some non-EU countries, the reported data are found to be partially available or not available at all. In other cases, the quality of the reported data is found to be insufficient, i.e. with important data gaps or following different formats or methods. In this case, emission data from the IIASA GAINS model instead (IIASA, 2018) are used. This model is the main tool used to underpin pan-European and EU level air quality policies such as the UNECE Convention on Long Range Transboundary Air Pollution (UNECE, 2012) and the EU National Emission reduction Commitments Directive (European Commission, 2016).

After collecting all the emission data from reporting and GAINS, the source sectors are harmonised, distinguishing around 250 different subsectors. For each detailed sector, a speciation is applied to the $PM_{2.5}$ and $PM_{10}$ emissions, distinguishing elemental carbon (representing BC in the HTAP_v3 inventory), organic carbon and other non-carbonaceous emissions for both the coarse (2.5-10 μm) and fine (<2.5 μm) mode.

A consistent spatial resolution is applied across the entire domain, where a specific proxy is selected for each subsector to spatially distribute emissions, including for instance the use of point source emissions, e.g., from the European Pollutant Release and Transfer Register (E-PRTR), complemented with additional data from the reporting of EU Large Combustion Plants (European Commission, 2001) and the Platts/WEPP commercial database for power plants (Platts, 2017). Road transport emissions are spatially disaggregated using information from OSM (Open Street Map, 2017), combined with information on traffic intensity in specific road segments from OTM (OpenTransportMap, 2017). Agricultural livestock emissions are spatially distributed using global gridded livestock numbers (FAO, 2010). Furthermore, CORINE land cover (Copernicus Land Monitoring Service, 2016) and population density are other key spatial distribution proxies.

After having spatially distributed the data, the ~ 250 different source categories are aggregated to fit with the HTAP_v3 sector classification (Table S1). CAMS-REG-v5.1 is an update of an earlier version, CAMS-REG-v4.2 and based on the 2020 submissions to cover the years 2000-2018. A detailed description of the CAMS-REG-v4.2 inventory is provided in Kuenen et al. (2022).

The data are provided as gridded annual totals at a resolution of 0.05°x0.1° (lat-lon). Along with the grids, additional information is available including height profiles as well as temporal profiles to break down the annual emissions into hourly data (monthly profiles, day-of-the-week profiles and hourly profiles for each day). Furthermore, the CAMS-REG inventory provides dedicated speciation profiles for NMVOC per year, country and sector.

**2.3.2 US EPA inventory**

Emissions estimates for the United States were based primarily on estimates produced for the EPA's Air QUAlity TimE Series Project (EQUATES), which generated a consistent set of modelled emissions, meteorology, air quality, and pollutant deposition for the United States spanning the years 2002 through 2017 (https://www.epa.gov/cmaq/equates). For each sector, a consistent methodology was used to estimate emissions for each year in the 16-year period, in contrast to the evolving methodologies applied in the triennial U.S. National Emissions Inventories (NEIs) produced over that span. The HTAPv3 time series was extended back one year to 2001 and forward one year to 2018 using country, sector, and pollutant specific trends from EDGARv6.1.

Emissions estimates were calculated for more than 8000 Source Classification Codes grouped into 101 sectors and then aggregated to the 16 HTAP_v3 emission sectors. The 2017 NEI (U.S. Environmental Protection Agency, 2021b) served as the base year for the time series. For each sector, emissions estimates were generated for previous years using one of four methods: 1) applying new methods to create consistent emissions for all years, 2) scaling the 2017 NEI estimates using annual sector-specific activity data and technology information at the county level, 3) using annual emissions calculated consistently in previous NEIs and interpolating to fill missing years, and 4) assuming emissions were constant at 2017 levels. The assumption of constant emissions was applied to a very limited number of sources. Foley et al. (2023) provides a detailed explanation of the assumptions used for each sector.

Emissions from electric generating units were estimated for individual facilities, combining available hourly emissions data for units with continuous emissions monitors (CEMs) and applying regional fuel-specific profiles to units without CEMS. On-road transport and non-road mobile emissions were estimated using emission factors from the MOVES v3 model (U.S. Environmental Protection Agency, 2021a). A complete MOVES simulation was completed only for the NEI years with national adjustment factors applied for years plus or minus one from the NEI year. For California, emission factors for all on-road sources for all years were based on the California Air Resources Board Emission Factor Model (EMFAC) (https://ww2.arb.ca.gov/our-work/programs/mobile-source-emissions-inventory/). New non-road emissions estimates for Texas were provided by the Texas Commission on Environmental Quality. Emissions from oil and gas exploration and production were calculated using point source specific data and the EPA Oil and Gas Tool (U.S. Environmental Protection Agency, 2021b), incorporating year-specific spatial, temporal, and speciation profiles. Residential wood combustion estimates were developed with an updated methodology incorporated into the 2017 NEI and scaled backward to previous years using a national activity as a scaling factor. Solvent emissions were estimated using the Volatile Chemical Product (VCPy) framework of Seltzer et al. (2021). Emissions from livestock waste were calculated with revised annual animal counts to address missing data and methodological changes over the period. Emissions for

agricultural burning were developed using a new suite of activity data with the same methodology and input data sets from 2002 onwards. County-level estimates were only available for 2002 because activity data based on satellite information was not yet available. Emissions for forest wildfires, prescribed burns, grass and rangeland fires were also calculated in EQUATES but not included in the HTAP_v3 data. For EQUATES, fugitive dust emissions (e.g., unpaved road dust, coal pile dust, dust from agricultural tilling) were reduced to account for precipitation and snow cover by grid cell. For use in HTAP_v3, however, no meteorological adjustments (which decrease annual $PM_{10}$ emissions by about 75% on average) were applied to fugitive dust emissions. Wind-blown fugitive dust emissions are not included in the estimates for other regions in the HTAP_v3 mosaic

Non-point source emissions were allocated spatially based on a suite of activity surrogates (e.g. population, total road miles, housing, etc.), many of which are sector specific. The spatial allocation factors were calculated for the 0.1 degree grid used by EDGARv6.1 with no intermediate re-gridding. The spatial allocation factors for all sectors were held constant for the entire time series except for oil and gas sectors which were year-specific. Depending on the sector, either 2017-based or 2014-based surrogates were developed for the same sectors as in the EQUATES.

Emissions from the US EPA inventory were provided from 2002-2017 (Table 1). Emissions for the year 2018 were estimated applying country, sector and pollutant specific trends from EDGAR, as well as for years 2000 and 2001 to complete the entire time series. Table S1 provides an overview about the US EPA inventory sector mapping to the HTAP_v3 sectors.

**2.3.3 Environment and Climate Change Canada (ECCC) inventory**

The Canadian emissions inventory data were obtained from 2018-released edition of Canada's Air Pollutant Emissions Inventory (APEI) originally compiled by the Pollutant Inventories and Reporting Division (PIRD) of Environment and Climate Change Canada (ECCC) (APEI, 2018). This inventory contains a comprehensive and detailed estimate of annual emissions of seven criteria air pollutants ($SO_2$, NOx, CO, NMVOC, $NH_3$, $PM_{10}$, $PM_{2.5}$) at the national and provincial/territorial level for each year for the period from 1990 to 2016. The APEI inventory was developed based on a bottom-up approach for facility-level data reported to the National Pollutant Release Inventory (NPRI) (NPRI, 2017), as well as an in-house top-down emission estimates based on source-specific activity data and emissions factors. In general, methodologies used to estimate Canadian emissions are consistent with those developed by the U.S. EPA (EPA, 2009) or those recommended in the European emission inventory guidebook (EMEP/EEA, 2013). These methods are often further adjusted by PIRD to reflect the Canadian climate, fuels, technologies and practices.

To prepare emissions in the desired HTAP classification, the APEI sector emissions were first mapped to the United Nations Economic Commission for Europe (UNECE) Nomenclature for Reporting (NFR) categories, which involved dividing the sector emissions into their combustion and process components. The NFR categories were then mapped to the HTAP 16 sector categories provided in the sector disaggregation scheme guide. Table S1 provides an overview of ECCC sector mapping to the HTAP_v3 sectors.

The HTAP-grouped APEI inventory emissions files were further processed by the Air Quality Policy-Issue Response (REQA) Section of ECCC to prepare the air-quality-modelling version of inventory files in the standard format (i.e., FF10 format) supported by the U.S EPA emissions processing framework. To process emissions into gridded, speciated and total monthly values, a widely-used emissions processing system called the Sparse Matrix Operator

Kernel Emissions (SMOKE) model, version 4.7 (UNC, 2019) was used. As part of the preparation for SMOKE processing, a gridded latitude-longitude North American domain at 0.1 x 0.1 degree resolution was defined with 920 columns and 450 rows covering an area of - 142W to -50W and 40N to 85N. The point-source emissions in the APEI include latitude and longitude information so those sources were accurately situated in the appropriate grid cell in the Canadian HTAP gridded domain. However, to allocate provincial-level non-point source emissions into this domain, a set of gridded spatial surrogate fields was generated for each province from statistical proxies, such as population, road network, dwellings, crop distributions, etc. Over 80 different surrogate ratio files were created using the 2011 Canadian census data obtained from Statistics Canada website (https://www12.statcan.gc.ca/census-recensement/2011/index-eng.cfm) and other datasets, such as the Canadian National Road Network (https://open.canada.ca/data/en/dataset/3d282116-e556-400c-9306-ca1a3cada77f).

To map the original APEI inventory species to the HTAP's desired list of species, PM speciation profiles from the SPECIATE version 4.5 database (EPA, 2016) were used to calculate source-type-specific EC and OC emissions. As a final step in SMOKE processing, the monthly emissions values were estimated using a set of sector-specific temporal profiles developed and recommended by the U.S. EPA (Sassi, 2021). For the point sources the NPRI annually reported monthly emissions proportions were applied. Emissions for the years 2017 and 2018 were calculated by applying sector- and pollutant-specific trends from EDGAR.

**2.3.4 REASv3.2.1 inventory**

The Regional Emission inventory in ASia (REAS) series have been developed for providing historical trends of emissions in the Asian region including East, Southeast, and South Asia. REASv3.2.1, the version used in HTAP_v3, runs from 1950 to 2015. REASv3.2.1 includes emissions of $SO_2$, $NO_x$, CO, NMVOCs, $NH_3$, $CO_2$, $PM_{10}$, $PM_{2.5}$, BC, and OC from major anthropogenic sources: fuel combustion in power plant, industry, transport, and domestic sectors; industrial processes; agricultural activities; evaporation; and others.

Emissions from stationary fuel combustion and non-combustion sources are traditionally calculated using activity data and emission factors, including the effects of control technologies. For fuel consumption, the amount of energy consumption for each fuel type and sector was obtained from the International Energy Agency World Energy Balances for most countries and province-level tables in the China Energy Statistical Yearbook were used for China. Other activity data such as the amount of emissions produced from industrial processes were obtained from related international and national statistics. For emission factors, those without effects of abatement measures were set and then, effects of control measures were considered based on temporal variations of their introduction rates. Default emission factors and settings of country- and region-specific emission factors and removal efficiencies were obtained from scientific literature studies as described in Kurokawa and Ohara (2020) and references therein.

Emissions from road transport were calculated using vehicle numbers, annual distance travelled, and emission factors for each vehicle type. The number of registered vehicles were obtained from national statistics in each country and the World Road Statistics. For emission factors, year-to-year variation were considered by following procedures: (1) Emission factors of each vehicle type in a base year were estimated; (2) Trends of the emission factors for each vehicle type were estimated considering the timing of road vehicle regulations in each country and the ratios of vehicle production years; (3) Emission factors of each vehicle type during the target period were calculated using those of base years and the corresponding trends.

In REASv3.2.1, only large power plants were treated as point sources. For emissions from cement, iron, and steel plants, grid allocation factors were developed based on positions, production capacities, and start and retire years for large plants. Gridded emission data of EDGARv4.3.2 were used for grid allocation factors for the road transport sector. Rural, urban, and total population data were used to allocation emissions from the residential sector. For other sources, total population were used for proxy data.

For temporal distribution, if data for monthly generated power and production amounts of industrial products were available, monthly emissions were estimated by allocating annual emissions to each month using the monthly data as proxy. For the residential sector, monthly variation of emissions was estimated using surface temperature in each grid cell. If there is no appropriate proxy data, annual emissions were distributed to each month based on number of dates in each month.

Monthly gridded emission data sets at 0.25°x0.25° resolution for major sectors and emission table data for major sectors and fuel types in each country and region during 1950-2015 are available in text format from a data download site of REAS (https://www.nies.go.jp/REAS/). Table S1 provides an overview about the REASv3.2.1 sector mapping to the HTAP_v3 sectors.

More details of the methodology of REASv3.2.1 are available in Kurokawa and Ohara (2020) and its supplement. (Note that REASv3.2.1 is the version after error corrections of REASv3.2 of Kurokawa and Ohara (2020)). Details of the error corrections are described in the data download site of REAS.) Table S1 provides an overview about the REASv3.2.1 sector mapping to the HTAP_v3 sectors.

The MEIC inventory (http://meicmodel.org, 2021) is not currently included in the HTAP_v3 mosaic. Since the REAS inventory only includes emissions until 2015, the REAS-based HTAP_v3 mosaic is only complete until this year. Emissions beyond 2015 were extrapolated using trends derived from a combination of MEIC and EDGAR. To extend the Chinese emission estimates to most recent years, MEIC data were used to adjust sector and pollutant specific trend for China for the years 2016 and 2017 (refer to Table S2 for the mapping sectors of MEIC and HTAP_v3). Then, the 2018 data were calculated based on the 2015-2017 trend. For all the other countries belonging to the REAS domain, the emissions were extended beyond 2015 applying the sector-, country-, and pollutant-specific trends from EDGAR.

## 2.3.5 CAPSS-KU inventory

In the Republic of Korea, the National Air Emission Inventory and Research Center (NAIR) estimates annual emissions of the air pollutants CO, NOx, SOx, TSP, $PM_{10}$, $PM_{2.5}$, BC, VOCs, and $NH_3$ via the Clean Air Policy Support System (CAPSS). The CAPSS inventory is divided into four source-sector levels (high, medium, low and detailed) based on the European Environment Agency's (EEA) CORe Inventory of AIR emissions (EMEP/CORINAIR). For activity data, various national- and regional-level statistical data collected from 150 domestic institutions are used. For large point sources, emissions are estimated directly using real-time stack measurements. For small point, area and mobile sources, indirect calculation methods using activity data, emission factors, and control efficiency are used.

Even though CAPSS (Clean Air Policy Support System) has been estimating annual emissions since 1999, some inconsistencies exist in the time series because of the data and methodological changes over the period. For example, emissions of $PM_{2.5}$ were initiated from the year 2011 and not from 1999. Therefore, in the CAPSS emission inventory, $PM_{2.5}$ emissions were calculated from 2011, and post-2011 the $PM_{10}$ to $PM_{2.5}$ emission ratio was used to calculate the emissions from 2000 to 2010. These limitations make it difficult to compare and analyse

emissions inter-annually. To overcome these limitations, re-analysis of the annual emissions of pollutants was conducted using upgrades of the CAPSS inventory, such as missing source addition and emission factor updates.

The biomass combustion and fugitive dust sector emissions from 2000 to 2014 were estimated and added in the inventory, which are newly calculated emission sources from 2015. As for the on-road mobile sector, new emission factors using 2016 driving conditions were applied from the year 2000 to 2015. Since the emissions from the combustion of imported anthracite coal were calculated only from 2007, the coal use statistics of imported anthracite from 2000 to 2006 were collected to estimate emissions for those years.

After all the adjustments, a historically re-constructed emissions inventory using the latest emission estimation method and data was developed. Table S1 provides an overview about the CAPSS sector mapping to the HTAP_v3 sectors.

## 2.3.6 JAPAN inventory (PM2.5EI and J-STREAM)

The Japanese emission inventory contributing to the HTAP_v3 mosaic is jointly developed by the Ministry of the Environment, Japan (MOEJ) for emissions arising from mobile sources and by the National Institute of Environmental Studies (NIES) for estimating emissions from fixed sources.

The mobile source emissions data for the HTAP_5.1, 5.2, and 5.4 sectors are based on the air pollutant emission inventory named "PM2.5 Emission Inventory (PM2.5EI), https://www.env.go.jp/air/osen/pm/info.html). PM2.5EI has been developed for the years 2012 and 2015, while for 2018 is currently under development. Almost all anthropogenic sources are covered, but emissions from vehicles are estimated in particular detail based on JATOP (Shibata, 2021). The emission factor of automobiles is constructed by MOEJ as a function of the average vehicle speed over several kilometres in a driving cycle that simulates driving on a real road. Emission factors are organized by 7 types of vehicles, 2 fuel types, 5 air pollutants, and regulation years, and have been implemented since 1997 as a project of MOEJ. By using these emission factors and giving the average vehicle speed on the road to be estimated, it is possible to estimate the air pollutant emissions per kilometre per vehicle. The hourly average vehicle speed of trunk roads, which account for 70% of Japan's traffic volume, is obtained at intervals of several kilometres nationwide every five years, so the latest data for the target year is used. For narrow roads, the average vehicle speed by prefecture measured by probe information is applied. It is 20 km/h in Tokyo, but slightly faster in other prefectures. Starting emission is defined as the difference between the exhaust amount in the completely cold state and the warm state in the same driving cycle and is estimated by the times the engine started in a day. Chassis dynamometer tests are performed in a well-prepared environment, so for more realistic emissions estimates, temperature correction factor, humidity correction factor, deterioration factor, DPF regeneration factor, and soak time correction factor are used. In addition to running and starting emissions, evaporative emissions from gasoline vehicles and non-exhaust particles such as road dust (including brake wear particles) and tire wear particles are combined to provide a vehicle emissions database with a spatial resolution of approximately 1 km × 1 km (30° latitude, 45° longitude), and a temporal resolution of an hour by month, including weekdays and holidays.

Further improvements of Japanese road transport emissions may be available in future updates of the HTAP_v3 mosaic.

Emissions from stationary sources in Japan are derived from the emission inventory developed in the Japan's Study for Reference Air Quality Modelling (J-STREAM) model intercomparison project (Chatani et al., 2018; Chatani et al., 2020). In this emission inventory, emissions from stationary combustion sources are estimated by multiplying emission factors and activities including energy consumption, which is available in the comprehensive energy statistics. Large stationary sources specified by the air pollution control law need to report emissions to the government every three years. The emission factors and their annual variations were derived from the emissions reported by over 100,000 sources (Chatani et al., 2020). For fugitive VOC emissions, MOEJ maintains a special emission inventory to check progress on regulations and voluntary actions targeting 30% reduction of fugitive VOC emissions starting from 2000. VOC emissions estimated in this emission inventory are used. Emissions from agricultural sources are consistent with the emissions estimated in the national greenhouse gas emission inventory (Center for Global Environmental Research et al., 2022). Emissions of all the stationary sources are divided into prefecture, city, and grid (approximately 1 x 1 km) levels based on spatial proxies specific to each source. Emissions for the year 2018 were estimated applying sector- and pollutant-specific trends from EDGAR. Table S1 provides an overview about the Japanese inventory sector mapping to the HTAP_v3 sectors.

## 2.4 Gap-filling methodology with EDGARv6.1

EDGAR is a globally consistent emission inventory of air pollutant and greenhouse gases developed and maintained by the Joint Research Centre of the European Commission (https://edgar.jrc.ec.europa.eu/). The EDGAR methodology used to compute GHG and air pollutant emissions has been described in detail in several publications (Janssens-Maenhout et al., 2019; Crippa et al., 2018) and summarised here after. In EDGAR, air pollutant emissions are computed making use of international statistics as activity data (e.g., International Energy Balance data, Food and Agriculture Organisation statistics, US Commodity statistics, etc.), region- and/or country-specific emission factors by pollutant/sector and technology and abatement measures, following Eq. 1:

$$EM_{i(C,t,x)} = \sum_{j,k} AD_{i(C,t)} * TECH_{i,j(C,t)} * EOP_{i,j,k(C,t)} * EF_{i(C,t,x)} * (1 - RED)_{i,j,k(C,t,x)}$$

(Eq. 1)

where EM are the emissions from a given sector i in a country C accumulated during a year t for a chemical compound x, AD the country-specific activity data quantifying the human activity for sector i, TECH the mix of j technologies (varying between 0 and 1), EOP the mix of k (end-of-pipe) abatement measures (varying between 0 and 1) installed with a share k for each technology j, and EF the uncontrolled emission factor for each sector i and technology j with relative reduction (RED) by abatement measure k. Emission factors are typically derived from the EMEP/EEA Guidebooks (EMEP/EEA, 2013, 2019, 2016), the AP-42(EPA, 2009) inventory and scientific literature.

Annual country and sector specific air pollutant emissions are then disaggregated into monthly values (Crippa et al., 2020) and subsequently spatially distributed making use of detailed proxy data (Janssens-Maenhout et al., 2019; Crippa et al., 2021).

As the most comprehensive and globally consistent emission database, the latest update of the EDGAR air pollutant emissions inventory, EDGARv6.1 (https://edgar.jrc.ec.europa.eu/dataset_ap61), is used in the HTAP_v3 mosaic to complete missing information from the officially reported inventories, as reported in Table 3. EDGARv6.1 includes important updates to estimate air pollutant emissions such as the improvement of road transport emission estimates for many world regions, the inclusion of agricultural NMVOC emissions, revised monthly emission profiles (Crippa et al., 2020) and updated spatial proxies to distribute national emissions by sector over the globe (Crippa et al., 2021). EDGARv6.1 also includes new international shipping proxies and their monthly distribution based on the STEAM model (Jalkanen et al., 2012; Johansson et al., 2017). In the Supplementary Material (section S2), the assessment of EDGAR emission data is reported in comparison with global and regional inventories.

**3 Results**

**3.1 Annual time series analysis: trends and regional and sectoral contributions**

Having a consistent set of global annual emission inventories for a two-decade period allows the investigation of global emissions trends for the inventory pollutants and regional and sectoral contributions. Figure 2 presents annual time series (2000-2018) of the global emissions of the nine air pollutants included in the HTAP_v3 mosaic separated into the actual contributions of 12 regions. Figure 3 shows the corresponding relative contributions of (a) 16 sectors and (b) 12 regions to the 2018 global emissions of these same pollutants. We can then discuss each pollutant in turn. In the following paragraphs we shortly present global and regional air pollutant emissions and their trends over the 2000-2018 period as provided by the HTAP_v3 data. Emissions are not presented with a confidence level since no comprehensive bottom-up uncertainty analysis has been performed in the context of the mosaic compilation, however see discussion in section 3.5.

Global $SO_2$ emissions declined from 99.4 to 72.9 Mt over the past two decades. This decreasing pattern is found for several world regions with the fastest decline in Eastern Asia, where after the year 2005 $SO_2$ emissions began to decrease steadily. This is consistent with the use of cleaner fuels with lower sulphur content and the implementation of desulphurisation techniques in power plants and industrial facilities in China in accordance with the 11th Five-Year Plan (FYP, 2006–2010 (Planning Commission, 2008)) and the 12th Five-Year Plan (FYP, 2011–2015 (Hu, 2016)) (Sun et al., 2018). Similarly, industrialised regions, such as North America and Europe, are characterised by a continuous decreasing trend in $SO_2$ emission, which had started well before the year 2000 due to the implementation of environmental and air quality legislation (EEA, 2022). Increasing $SO_2$ emissions, on the other hand, are found for Southern Asia (+112% compared to 2000), South-East Asia and developing Pacific (+62%), and Africa (+40%). These increases mostly arise from the energy, industry, and (partly) residential sectors, and reflect the need for emerging and developing economies to mitigate these emissions. Emissions estimated using satellite retrievals and model inversions confirm the trends provided by the HTAP_v3 mosaic (Liu et al., 2018). $SO_2$ is mostly emitted by power generation and industrial activities, which in 2018 represent 42% and 26%, respectively, of the global total. Despite measures in some specific sea areas to mitigate sulphur emissions, globally they have been rising steadily with increasing activity. International shipping represents 13.8% of global $SO_2$ emissions in 2018, and it is 41% higher compared to the 2000 levels (Fig. 3).

Global NOx emissions increased from 110.4 Mt in 2000 to 117.4 Mt in 2018 as a result of the increase in energy- and industry-related activities for most of the world regions (in particular over the Asian domain). The strongest decreases are found for North America (-63%), Europe (-42%), Asia-Pacific Developed (-32%) and to a lower extent for Eurasia (-6%). Comparable spatio-temporal patterns are found by satellite OMI data and ground based measurements of $NO_2$ concentrations (Jamali et al., 2020). NOx is mainly produced at high combustion temperatures (e.g., power and industrial activities, 38% of the global total), but also by transportation (27% of the global total) and international shipping (14% of the global total).

CO is mostly emitted by incomplete combustion processes from residential combustion, transportation and the burning of agricultural residues. Globally, CO emissions showed little change over the past two decades (502.7 Mt in 2000 vs. 499.8 in 2018), but different regional trends are present. Historically industrialised regions have reduced their emissions over the years (-42% in Europe and -62% in North America), while CO emissions increased in Africa by 45% and in Southern Asia by 49%. Road transport CO emissions halved over the past two decades (-55%), while the emissions from all other sectors increased. These results are consistent with MOPITT satellite retrievals, which mostly show the same trends over the different regional domains over the past decades (Yin et al., 2015).

NMVOC emissions increased from 115.2 Mt in 2000 to 146 Mt in 2018. These emissions are mostly associated with the use of solvents (25% of the 2018 global total), fugitive emissions (23%), road transportation (including both combustion and evaporative emissions, 15%) and small-scale combustion activities (19%). The most prominent increases in the emissions at the global level are found for the solvents sector (+81%). In 2018, NMVOC emissions from solvents were 3.7 and 3.5 times higher than in 2000 in China and India, respectively, while a rather stable trend in found for US and Europe.

Global $NH_3$ emissions increased from 47.4 Mt in 2000 to 58.9 Mt in 2018 due to enhanced emissions from agricultural activities. In particular, NH3 emissions strongly increased in Africa (+60.5%), South-East Asia and developing Pacific (48.3%), Southern Asia (+38.7%), and Latin America and Caribbean (+41.1%).

Particulate matter emissions showed little change over the past two decades at the global level, whereas regional emission increases are found for Africa (e.g., +47.0% for $PM_{10}$), Latin America and Caribbean (+39.1%), Middle East (48.3%), and Southern Asia (+56%), mostly associated with increases in agricultural waste burning and the livestock, energy, and waste sectors. By contrast, Eastern Asia (-39.5%), Europe (-24.3%), and Asia-Pacific Developed (-36.8%) significantly decreased their PM10 emissions over the past two decades due to the continuous implementation of reduction and abatement measures for the energy, industry, road transport and residential sectors (Crippa et al., 2016). As shown in Fig. 3, the relative contribution of North America to global $PM_{10}$ is quite high compared to other substances due to fugitive dust emissions (e.g., unpaved road dust, coal pile dust, dust from agricultural tilling) which have not been adjusted for meteorological conditions (e.g., rain, snow) and near-source settling and mitigation (e.g., tree wind breaks) because these removal mechanisms are better addressed by the chemical transport models. Additional uncertainty may be therefore introduced for these emissions, depending on the modelling assumptions of each official inventory. Similarly, particulate matter speciation into its carbonaceous components is often challenging and subjected to higher level of uncertainty, for instance because different definitions are used for PM in inventories, including condensable emissions or not (Denier van der Gon et al., 2015). Attempts to improve the accuracy of such emissions (e.g. BC and OC emissions over the European domain) are ongoing.

Figure 3a shows more than 50% difference at the global level between $PM_{2.5}$ emissions and the sum of its carbonaceous components (BC and OC), which however varies depending on the region and sector. The largest difference between $PM_{2.5}$ and the sum of BC and OC is generally found for the energy and industrial sectors, where due to the high temperatures BC and OC are largely burned. Within this sector, the non carbonaceous fraction of $PM_{2.5}$ represents around 75% in Europe, 78% in the USA and up to more than 95% over Asian countries (e.g. China and India). This PM fraction is represented by other minerals, ash (mostly when burning coal) and sulphate. Road transport is also a sector showing large differences between $PM_{2.5}$ and the sum of BC and OC, with around 40% difference for Europe, around 90% difference for USA and lower values for India and China (around 15%). This component may be associated with other minerals. For the residential sector, this difference is generally lower and around 25% (for Europe and Asian countries), while around 37% in the USA and is possibly associated with other minerals and ash due to coal combustion. Shipping is also a sector where a large component of $PM_{2.5}$ (around 70%) is not associated with carbonaceous fractions but to sulphate. In particular, regions within the Sulphur Emission Control Area (SECA) show lower contributions from sulphates (e.g. Europe and USA) with an overall contribution of 5-10%. Another source of uncertainty which may contribute to enhancing the difference between $PM_{2.5}$ and the sum of BC and OC is associated on how different inventories consider condensable particulate matter.

**3.2 Emission maps**

Spatially distributed emission data describe where emissions take place, as input for local, regional and global air quality modelling. As noted in section 2.2, nationally aggregated air pollutant emissions are spatially distributed over the corresponding national territory using spatial proxy data which are believed to provide a relatively good representation of where emissions takes place. Depending on the emitting sector, air pollutants can be associated with the spatial distributions of point sources (e.g., in the case of power plant or industrial activities), road networks (e.g., for transportation related emissions), settlement areas (e.g., for small-scale combustion emissions), crop and livestock distribution maps, ship tracks etc. Using reliable and up-to-date spatial information to distribute national emissions is therefore relevant, although challenging. Multiple assumptions are often made by inventory compilers when developing their inventories, which may result in differences when analysing spatially distributed emissions provided by different inventory compilers over the same geographical domain.

One key goal of the HTAP_v3 mosaic is to collate in one inventory the most accurate spatially-distributed emissions for all air pollutants at the global level, based on the best available local information. Point sources related with emissions from power plant and industrial facilities represent one the most critical spatial information to be retrieved, and their misallocation can significantly affect the characterisation of local air quality. This challenge is also present in the HTAP_v3 mosaic. For example, the REASv3.2.1 inventory is still using limited information to distribute emissions from these two sectors especially for industrial plants. Depending on the region, point source information could be limited compared to datasets used in inventories of North America and Europe. To overcome this issue, the participation of national emission inventory developers not only from China, but also India and other Asian countries is recommended. The impact can be seen in Fig. 4, which shows the global map of $SO_2$ emissions in 2018 based on the HTAP_v3 mosaic compilation, where information about the magnitude and the type of emission sources for the different regions can be retrieved. The energy and industry sectors contribute a large fraction of $SO_2$ emissions (Fig. 3a), but the spatial distribution of these emissions is qualitatively different in North America and Europe than in

Asia (i.e., more "spotty", less smooth and widely distributed). Ship tracks cover the entire geographical marine domain, consistent with emissions from the STEAM model (Jalkanen et al., 2012; Johansson et al., 2017) included in the EDGARv6.1 database, although showing marked emissions over the Mediterranean Sea, Asian domain, Middle East and North American coasts. Furthermore, emissions from power plant and industrial activities, as well as small-scale combustion are prominent over the Asian domain, Eastern Europe, and some African regions.

Sector-specific case studies are presented in the maps of Figs. 5-8. Figure 5 shows the comparison of annual NOx emissions for the year 2000 and 2018. The road transport sector is a key source of NOx emissions (cf. Fig. 3a), and this contribution is reflected in the visible presence of road networks in the maps. Decreasing emissions are found for industrialised regions (USA, Europe, Japan) thanks to the introduction of increasingly restrictive legislation on vehicle emissions since the 1990s, whereas a steep increase is found for emerging economies and in particular India, China, and the Asian domain. Figure 6 shows the different spatial allocation of $PM_{10}$ emissions from the residential sector during the month of January 2018, with higher emission intensities evident in the Northern Hemisphere (cold season) and the lower values in the Southern Hemisphere (warm season). Figures 7 and 8 show the spatio-temporal allocation of agriculture-related emissions, and specifically, $PM_{10}$ emissions from agricultural waste burning and $NH_3$ emissions from agricultural soil activities.

**3.3 Monthly temporal distribution**

**3.3.1 Monthly variability by region**

The magnitude of air pollutant emissions varies by month because of the seasonality of different anthropogenic activities and their geographical location (e.g., Northern vs. Southern Hemisphere regions). Figures 9 and 10 (and S3.1, S3.2 and S3.3) show the monthly distribution of regional emissions for those pollutants and sectors for which higher variability is expected. The year 2015 was chosen since it is the last year for which all of the official data providers have data. Figure 9 shows monthly $NH_3$ emissions by region from three agricultural activities (agricultural waste burning, livestock, and crops). These sectors display the largest variability by month, reflecting the seasonal cycle and the region-specific agricultural practices, such as fertilisation, crop residue burning, manure and pasture management, animal population changes, etc. In Figure 10, NOx emissions from residential activities show a particular monthly distribution, with the highest emissions occurring during the cold months shifted for the Northern and Southern Hemispheres. By contrast, regions in the equatorial zone do not show a marked monthly profile even for residential activities. The energy sector also follows monthly-seasonal cycles related to the demand for power generation, which is also correlated with ambient temperature and local day length. Transport-related emissions do not show a large variation by month, whereas daily and weekly cycles for transport-related emissions, which are typically more relevant, are beyond the temporal resolution of this work.

Although a spatio-temporal variability of the HTAP_v3 emissions is found in these figures, a more in-depth analysis reveals that with the exception of few regions and sectors (e.g., Canada, USA and regions gap-filled with EDGAR), no inter-annual variability of the monthly profiles is present, meaning that the majority of official inventories assume the same monthly distribution of the emissions for the past two decades (refer to Figs. S3.4-S3.9). This is different from the approach used for example by EDGAR (Crippa et al., 2020), ECCC for Canada, and U.S. EPA for the USA, where year-dependent monthly profiles are used for specific sectors, in particular for residential, power generation, and agricultural activities. Further analysis has shown that for the European domain regional rather than country-specific monthly profiles are

applied. Therefore, for Europe new state-of-the-art profiles have been made available under the CAMS programme by Guevara et al. (2021).

### 3.3.2 Spatially-distributed monthly emissions

An important added value of HTAP_v3 comes from the availability of monthly gridmaps that reflect the seasonality of the emissions for different world regions. Access to spatially distributed monthly emissions is essential to design effective mitigation actions, providing information on hot spots of emissions and critical periods of the year when emissions are highest.

Figure 11 shows mid-season $PM_{2.5}$ monthly emissions arising from the residential sector in 2018. The global map shows higher emissions in the Northern Hemisphere during January, while the opposite pattern is found for the Southern Hemisphere in July. Agriculture is an important activity characterised by strong seasonal patterns, as shown in Figs. 12 and 13. Figure 12 shows $PM_{10}$ monthly emission maps from agricultural residue burning in 2018 from HTAP_v3, highlighting higher emissions over certain months of the year related with specific burning practices of agricultural residues for different world regions. For example, during the month of April, intense burning of crop residues is found in Africa (Nigeria, Ethiopia, Sudan, South Africa, etc.), South America (Brazil, Argentina, Colombia, etc.), Northern India, and South-Eastern Asia (e.g., Vietnam, Thailand, Indonesia, Philippines, etc.). Figure 13 represents the yearly variability of $NH_3$ emissions from agricultural soils activities, mostly related with fertilisation. During the month of March and April, intense agricultural soils activities are found over Europe and North America compared to other months, while during the month of October the highest emissions are for this sector are found in China, India, several countries of the Asian domain, but also in USA, Australia, and Latin America. These results are consistent with satellite based observations performed using Cross-track Infrared Sounder (Shephard et al., 2020).

### 3.4 Vertical distribution of the emissions

### 3.4.1 Aircraft emissions

In EDGAR6.1 the emissions are provided at three effective altitude levels (landing/take-off, ascent/descent, and cruising). The spatial proxy for the aviation sector is derived from International Civil Aviation Organization (ICAO, 2015) which specifies a typical flight pattern with landing/take-off cycle within few km of the airport, followed by climb-out/descending phase during the first 100 km and the last 100km of a flight and finally the remaining part from 101 km until the last 101 km as the cruise phase. Routes and airport locations are taken from the Airline Route Mapper of ICAO (2015). In HTAP_v3, aircraft emissions are provided as domestic and international, but with no information about altitude ranges. We recommend modellers to use the corresponding EDGARv6.1 data (https://edgar.jrc.ec.europa.eu/dataset_ap61) including the vertical distribution of the emissions.

### 3.4.2 Speciation of NMVOC emissions

For emission data to be useful for modellers, total NMVOC emissions must be decomposed into emissions of individual NMVOC species. As the chemical mechanisms used by models can differ with respect to the NMVOC species they include, it is not practical to provide an NMVOC speciation which is usable by all models. Instead, a speciation is provided here for

the set of 25 NMVOCs defined by Huang et al. (2017) and the corresponding data are made available on the HTAP_v3 website. The absolute values of 25-category speciated NMVOC emissions were obtained for all countries for the 28 EDGAR sectors from here: *https://edgar.jrc.ec.europa.eu/dataset_ap432_VOC_spec*. The absolute NMVOC emissions of each species from each sector in this dataset were remapped to the HTAP_v3 sectors following the mapping from Table 2, then converted to a speciation by dividing by the total emissions of each individual species for the four world regions defined by Huang et al. (2017): Asia; Europe; North America; and Other. The resulting NMVOC speciation is provided in the supplementary material to this paper for the 25 NMVOC species, 4 world regions, and 15 emitting NMVOC sectors[2] following the HTAP_v3 sector classification (including 13 sectors defined over the 4 world regions, and the two international sectors: international shipping and international aviation). The list of countries comprising each region is also provided in the supplement.

**3.5 Emission Uncertainties**

**3.5.1 Overview on uncertainties**

Unlike greenhouse gas inventories, uncertainty is not routinely estimated for air pollutant emissions by country inventory systems. In part this is due to the different and often disparate processes used to generate air pollution data at the country level (Smith et al., 2022), making it more difficult to conduct uncertainty analysis. While combinations of observational and modelling techniques can be used to evaluate air pollutant emissions, these are inherently site specific and can be difficult to generalize.

The potential level of uncertainty in any emission estimate depends on how much emission factors vary for a particular activity. We note that the emission species with the lowest uncertainty is carbon dioxide from fossil fuel combustion. This is because $CO_2$ emission factors are closely tied to fuel energy content, which is a quantity that is tracked and reported by both government and commercial reporting systems. Similar considerations apply to $SO_2$ emissions, where emissions can be reliably estimated if the sulphur content of fuels and the operational characteristics of emission control devices are known. A key aspect here is that uncertainty in fuel sulphur content is largely uncorrelated across regions, which means that global uncertainty is relatively low, while regional uncertainty often much higher (Smith et al., 2011). On the opposite end of the spectrum, the emission rates for particulate matter depend sensitively on combustion conditions and the operation of any emission control devices and can vary over several orders of magnitude. While this is not an indication of the uncertainty in inventory estimates, this indicates the difficulty of constructing quantitative uncertainty estimates. The type of emission process also influences uncertainty, with fugitive emissions and emissions associated with biological processes generally having higher uncertainty levels.

We note also that uncertainty in the overall magnitude of emissions does not necessarily imply a similar level of uncertainty in relative emission trends. Even with uncertainties, the widespread use of emission control devices has resulted in reductions in air pollutant emissions in North America and Europe (Liu et al., 2018; Jamali et al., 2020), as verified by observational and modelling studies.

The emissions in the HTAP_v3 mosaic emissions originate from a variety of sources which has some implications for relative uncertainty. Emissions for some regions, such as North

---

[2] No speciation profile is provided for the 'tyre and brake wear sector' not being a source of NMVOC emissions.

America and Europe, were generated by country inventory systems which have been developed and refined over the last several decades. It is reasonable to assume these emissions are robust, however even in these regions detailed studies have indicated that actual emissions in some cases appear to be lower than inventory values (Anderson et al., 2014; Hassler et al., 2016; Travis et al., 2016). Where EDGAR emission estimates were used in the mosaic uncertainties are likely be higher overall given that inventory information developed in those countries was not available for these regions (Solazzo et al., 2021).

Some information on the robustness of the HTAP_v3 mosaic can be gained by comparing different inventory estimates, which is shown in supplement section S2. In many cases, the agreement between estimates (for example in North America and Europe) simply indicates common data sources and assumptions, although this does indicate that the different inventory groups did conclude that these values were plausible. The larger differences in other regions, however, does point to larger uncertainty there.

### 3.5.2 Qualitative assessment of the uncertainty of a global emission mosaic

Assessing the uncertainty of a global emission mosaic is challenging since it consists of several bottom-up inventories and by definition it prevents a consistent global uncertainty calculation. Each emission inventory feeding the HTAP_v3 mosaic is characterized by its own uncertainty which is documented, where available, by the corresponding literature describing each dataset (see Table 2 and section 2.3). However, the mosaic compilation process may also introduce additional uncertainties compared to the input datasets. In order to limit these additional uncertainties, we made the following considerations:

-for each emission inventory both the national totals and gridded data by sector were gathered. This process allows the mosaic compilers not to introduce additional uncertainty compared to the original input regional datasets. While additional uncertainties may arise from the extraction of the national totals from spatially distributed data (e.g. country border issues which were one limitation of previous editions of the HTAP mosaics), this is not the case in the current dataset. Therefore, when regional trends are described by region and pollutant (see section 3), no additional source of uncertainty has to be considered from the mosaic compilation approach.

-the sector definition and mapping has been developed following the IPCC categories and when no data was available for a certain combination of sector and pollutant a gapfilling procedure is applied using the EDGAR database. Therefore, the datasets are comparable in terms of sectoral coverage, which reduces uncertainties in this aspect.

- since each inventory provided monthly resolution emission gridmaps and time series there is no additional uncertainty introduced by temporal disaggregation as part of the construction of the HTAP_v3 mosaic.

In this work we also provide a qualitative indication of the emission variability by HTAP sector and pollutant at the global level. Table S6 summarises the variability of global HTAP_v3 emissions by sector for the boundary years of this mosaic (2000 and 2018) compared to the global EDGARv6.1 data. EDGAR emissions are considered as the reference global emission inventory against which comparing the HTAP_v3 estimates although these two global products are not fully independent. The variability of the global emissions is calculated as the relative difference of the estimates of the two inventories, i.e. (EDGARv6.1-HTAP_v3)/HTAP_v3). Emission variabilities are also classified as low (L, L<15%), low medium (LM,

15%<LM<50%), upper medium (UM, 50%<UM<100%), high (H, H>100%), based on the EMEP/EEA Guidebook (2019) information. The largest variability is found domestic shipping emissions (CO and NMVOC), energy (OC, BC), agricultural crops (PM), road transport (PM, NMVOC) and industry (NH3, NMVOC). In absence of a full uncertainty assessment the variability can be used as proxy of structural uncertainty, keeping in mind that variability could be biased towards overconfidence, thus underestimating the uncertainty. Furthermore, the uncertainty of the spatial proxies has not been assessed and maybe subject of future activity updates.

## 4 Data availability

The HTAP_v3 emission mosaic data can be freely accessed and cited using https://doi.org/10.5281/zenodo.7516361. All data can be also accessed through the EDGAR website at the following link: https://edgar.jrc.ec.europa.eu/dataset_htap_v3.

Data are made available in the following formats:

- Monthly gridmaps of emissions (in Mg/month) at 0.1x0.1degree resolution: there is one .NetCDF file per year and substance that includes the emissions for each sector for the 12 months.
- Monthly gridmaps of emission fluxes (in kg/m2/s) at 0.1x0.1degree resolution: there is one .NetCDF file per year and substance that includes the emission fluxes for each sector the emission fluxes for the 12 months.
- Annual gridmaps of emissions (in Mg/year at 0.1x0.1degree resolution: there is one .NetCDF file per year and substance that includes the emissions for each sector.
- Annual gridmaps of emission fluxes (in kg/m2/s) at 0.1x0.1degree resolution: there is one .NetCDF file per year and substance that includes the emission fluxes for each sector.

The full set of HTAP_v3 data is quite large, requiring substantial network bandwidth and time for download, and substantial storage space. To make it easier for users to query and use the data, additional products are available. For global modellers who may not require such high spatial resolution, gridmaps at 0.5x0.5 degree resolution are made available following the abovementioned specifications of the higher spatial resolution data. Furthermore, to allow regional modellers to download only the data for the regions they need, the JRC EDGAR group has also developed an interface to allow the users of the HTAP_v3 mosaic to extract emission data over arbitrarily specified geographical domains. The HTAP tool is accessible after creation of an ECAS account (https://webgate.ec.europa.eu/cas/login) and it is available at: https://edgar.jrc.ec.europa.eu/htap_tool/.

## 5 Conclusions

The global air pollution mosaic inventory HTAP_v3 presented and discussed in this paper is a state-of-the-art database for addressing the present status and the recent evolution of a set of policy-relevant air pollutants. The inventory is made by the harmonization and blending of six regional inventories, gapfilled using the most recent release of EDGAR (EDGARv6.1). By directly incorporating the best available local information, including the spatial distribution of emissions, the HTAP_v3 mosaic inventory can be used for policy-relevant studies at both

regional and global levels. As such, the HTAP_v3 mosaic inventory provides a complement to globally consistent emission inventories such as EDGAR. The global and regional trends of air pollutant emissions in the HTAP_v3 mosaic are comparable with other commonly available global emission datasets.

By providing consistent times series for almost two decades, HTAP_v3 allows an evaluation of the impact and success of the pollution control measures deployed across various regions of the world since 2000. Similarly, its finer sectoral resolution is suitable for understanding how and where technological changes have resulted in emissions reductions, suggesting possible pathways for strengthening appropriate policy actions.

All these features make HTAP_v3 a database of interest for policy makers active in the air quality regulatory efforts. HTAP_v3 provides a picture of a world where most pollutant emissions are following a steady or decreasing path. However, several areas of the world show an increasing emission trend, with wide portions of the world remaining subjected to unsatisfactory levels of ambient air quality.

When using the HTAP_v3 emission mosaic, users should consider the following limitations, for example when combining the HTAP_v3 data with other emission input needed to run atmospheric models:

- agricultural waste burning emissions should be treated with caution to avoid double-counting when combined with existing biomass burning emission inventories;

- NMVOC and NOx emissions from agricultural soils should be treated with caution to avoid double-counting when combining the HTAP_v3 data with a natural emissions model such as MEGAN (Model of Emissions of Gases and Aerosols from Nature);

- the speciation of NOx emissions into its components (NO, NO2, HONO) is not provided by the global HTAP_v3 mosaic and it is beyond the scope of the current work since the regional inventories report total NOx with no speciation. Standard practice in global models is to emit all anthropogenic NOx as NO, while we expect that regional modelling groups will have access to appropriate best practices for their particular regions. In particular for road transport, the partitioning of NOx emissions between NO, NO2, and HONO is highly region-dependent and it is based on the fleet composition (e.g., number of diesel vehicles relative to gasoline vehicles) and technology level (e.g., the level of exhaust after treatment).

Thanks to the continuous improvement of local and regional emission inventories, recent literature shows new datasets that report regional information over areas of the world not covered by local inventories in the current HTAP_v3 mosaic (e.g. Argentina (Puliafito et al. 2021), Africa (Keita et al., 2021) and the MEIC inventory (http://meicmodel.org.cn/?page_id=1772&lang=en)). Future updates to this mosaic may also integrate reliable and up to data information over South America or Africa as time and resources permit.

Similar to its predecessor (e.g. HTAP_v2.2 mosaic inventory), we expect that this new HTAP_v3 mosaic inventory will be used as a basis for global assessments of long-range, transboundary transport of air pollution under the Task Force on Hemispheric Transport of Air Pollution, while also providing a convenient and useful information for regional modellers seeking the best available regional emissions with a consistent gap-filling methodology.

## Author contributions.

MC and DG developed the mosaic gathering input from all data providers. The co-chairs of the TF-HTAP (TK, TB, RW and JaKa) fostered the dialogue with international institutions contributing to this work with their data. PM, RM, JR, JZ, DN, MS, MDM, RW provided data for Canada, JuKu, SC, TM provided data for Japan, JeKu provided data for Europe, J-HW, JK provided data for Korea, TK, GP provided data for USA, JiKi provided data for Asia. The JRC EDGAR group (MC, ES, DG, EP, MM, FM, ES, MB, FP) lead the drafting of the publication with input from colleagues contributing to the HTAP_v3 mosaic. SJS and HS performed detailed data comparison among available emission inventories. TA calculated and provided the NMVOC speciation fractions for all the sectors for the four regions.

**Competing interests.** The authors declare that they have no conflicts of interest nor competing interests.

**Acknowledgements.**

The authors would like to thank all HTAP_v3 data providers for the fruitful cooperation. The views expressed in this publication are those of the authors and do not necessarily reflect the views or policies of the European Commission.

J-STREAM emission inventory for Japan was developed by Environment Research and Technology Development Fund (JPMEERF20165001 and JPMEERF20215005) of the Environmental Restoration and Conservation Agency Provided by the Ministry of Environment of Japan, and FRIEND (Fine Particle Research Initiative in East Asia Considering National Differences) Project through the National Research Foundation of Korea (NRF) funded by the Ministry of Science and ICT (2020M3G1A1114622).

REASv3.2.1 has been supported by the Environmental Research and Technology Development Fund (grant and no. S-12 and S-20 (JPMEERF21S12012)) of the Environmental Restoration and Conservation Agency of Japan and the Japan Society for the Promotion of Science, KAKENHI (grant no. 19K12303)).

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

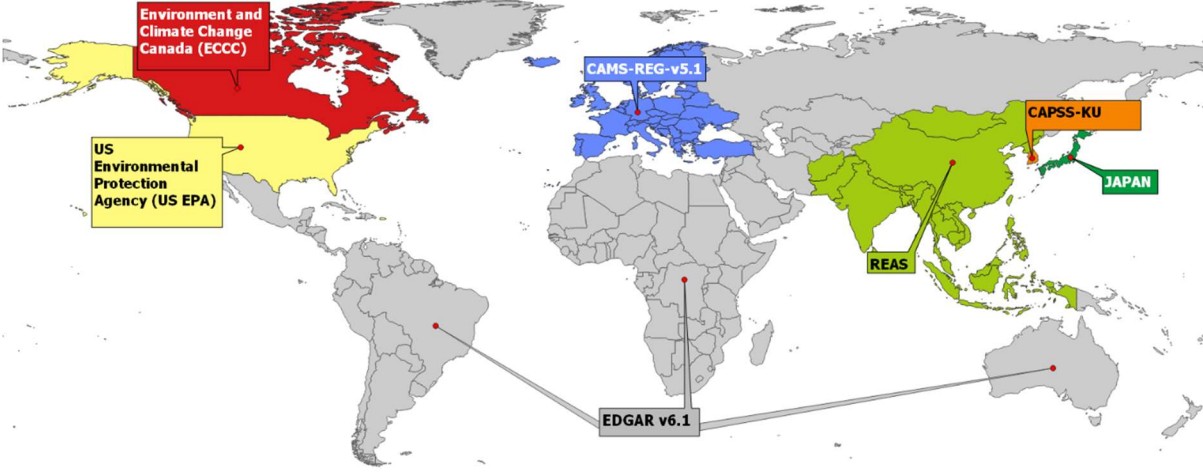

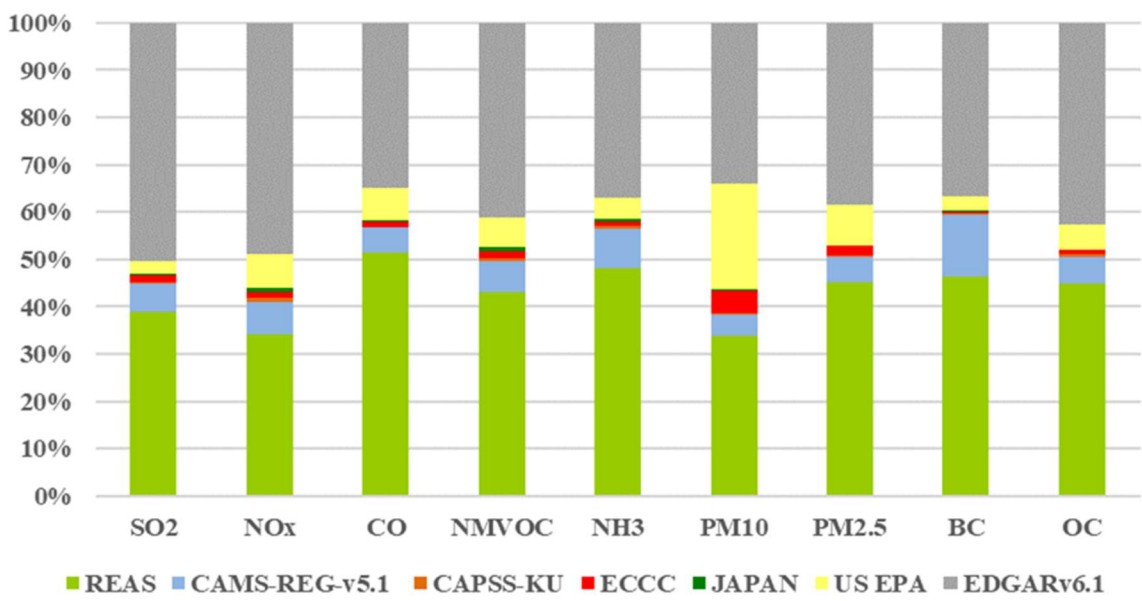

**Figure 1 – Overview of the HTAP_v3 mosaic data providers. Data from officially reported emission gridmaps were collected from the US Environmental Protection Agency, Environment and Climate Change Canada, CAMS-REG-v5.1 for Europe, REASv3.2.1 for most of the Asian domain, CAPSS-KU for Korea and JAPAN (PM2.5EI and J-STREAM) for Japan. The share of the total emissions covered by each data provider is reported in the bar chart at the bottom.**

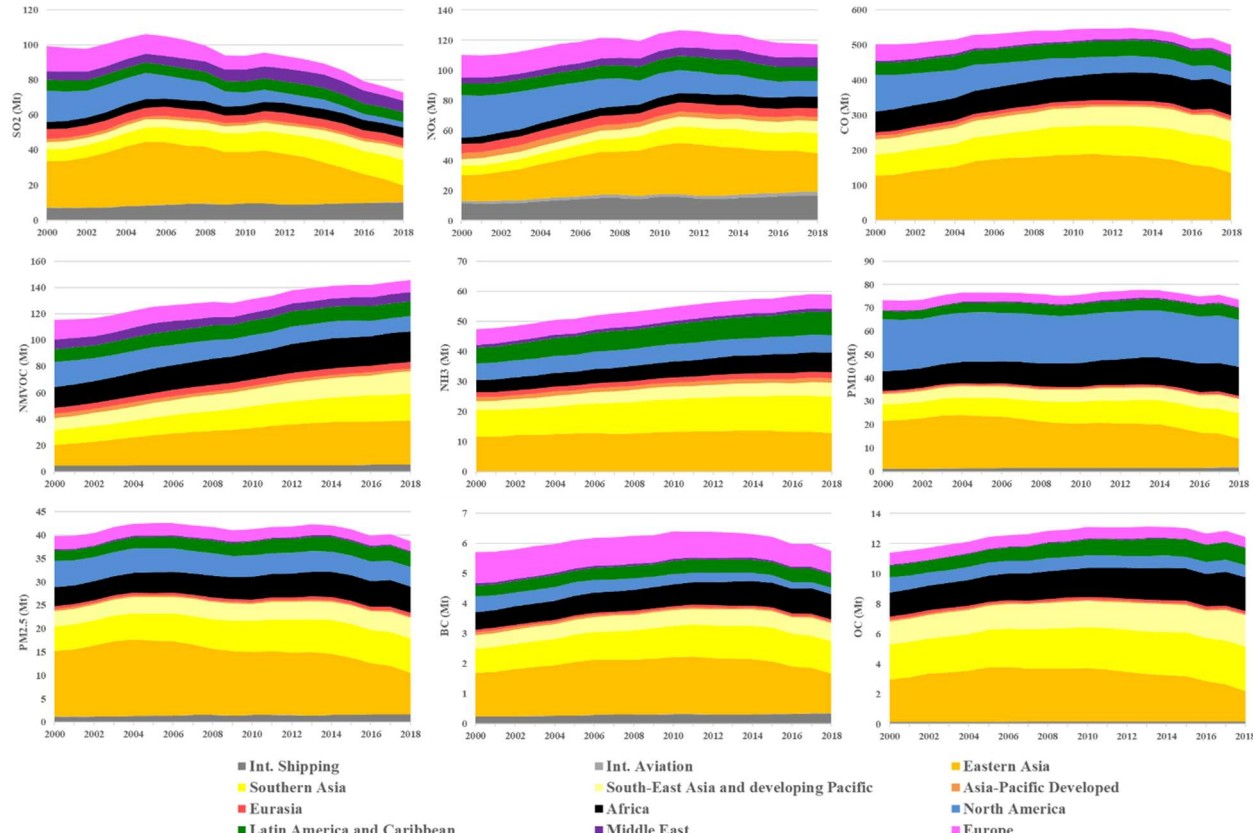

**Figure 2 – Time series of gaseous and particulate matter pollutants from HTAP_v3 by aggregated regions. Regional grouping follows the Intergovernmental Panel on Climate Change Sixth Assessment Report (IPCC AR6) definitions. Table S3 provides information on the regional belonging of each country to the IPCC AR6 regions.**

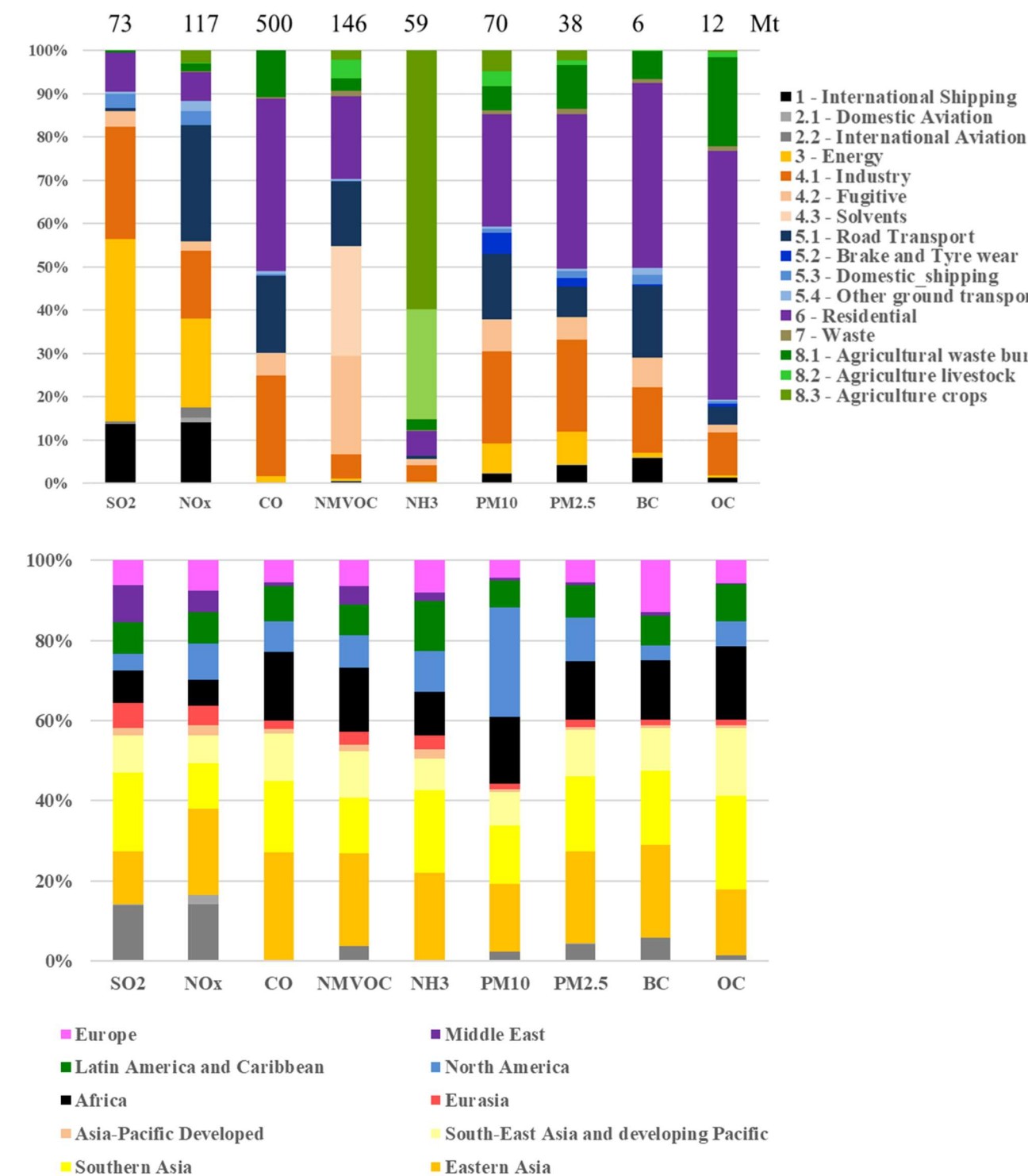

Figure 3 - Sectoral (panel a) and regional (panel b) breakdown of air pollutant emissions
from HTAP_v3 for the year 2018. At the top of each bar in panel a, total emissions for
each pollutant are reported (in Mt).

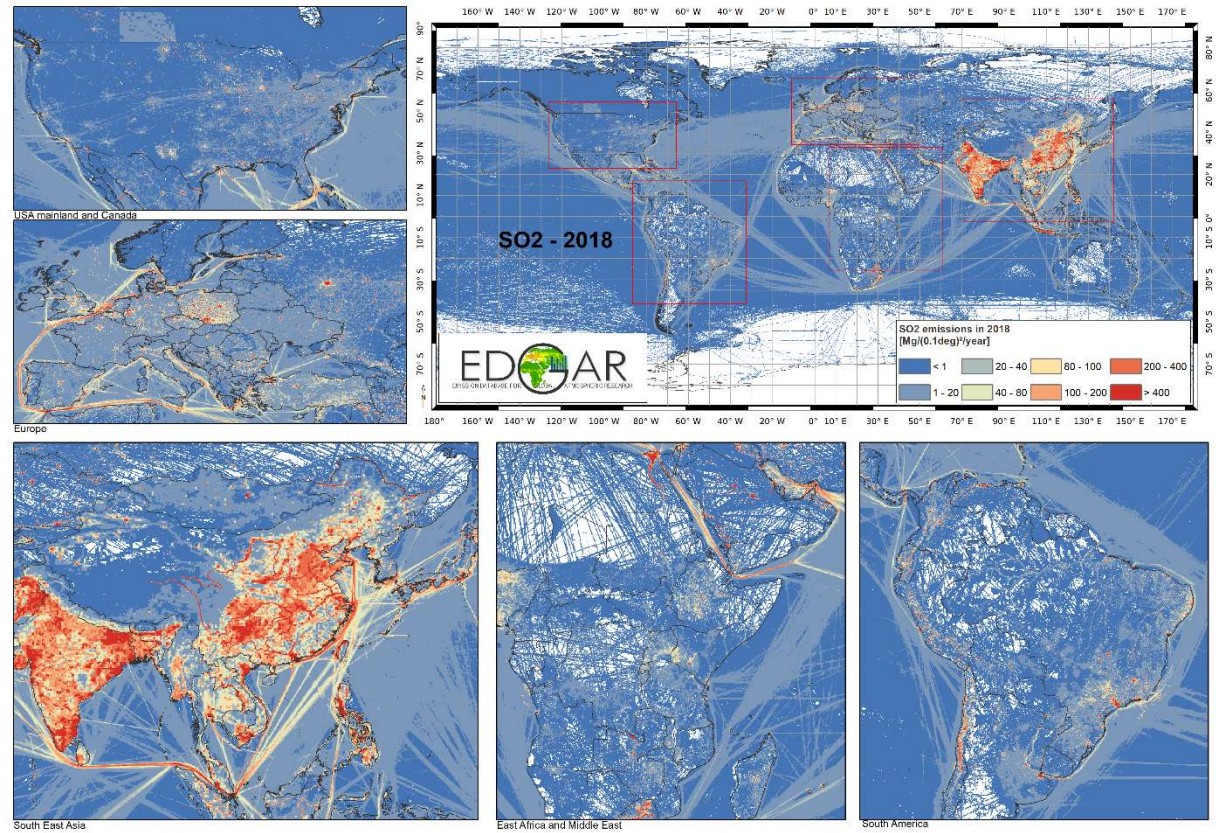

Figure 4 – HTAP_v3 mosaic: SO$_2$ emission gridmaps for the year 2018.

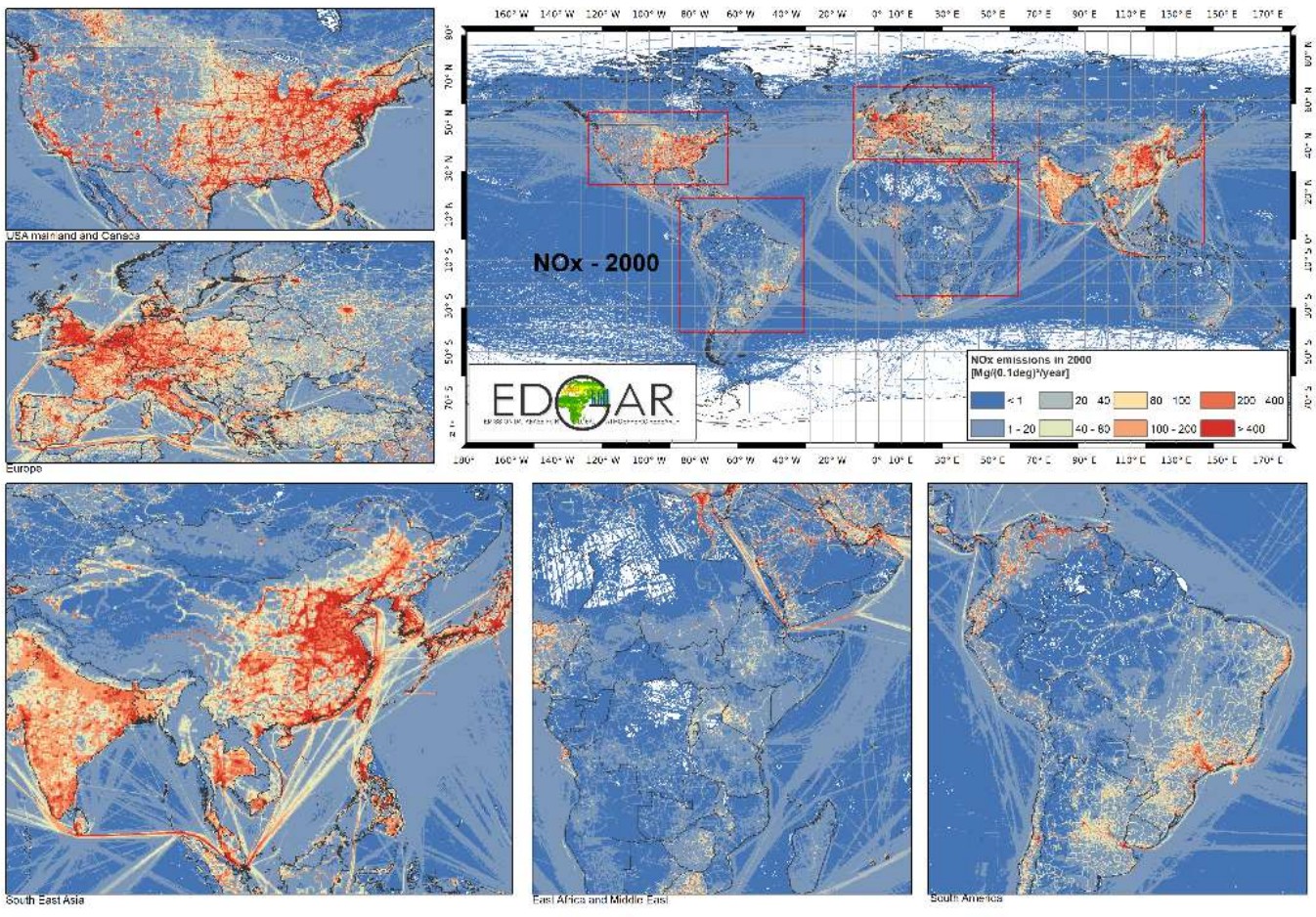

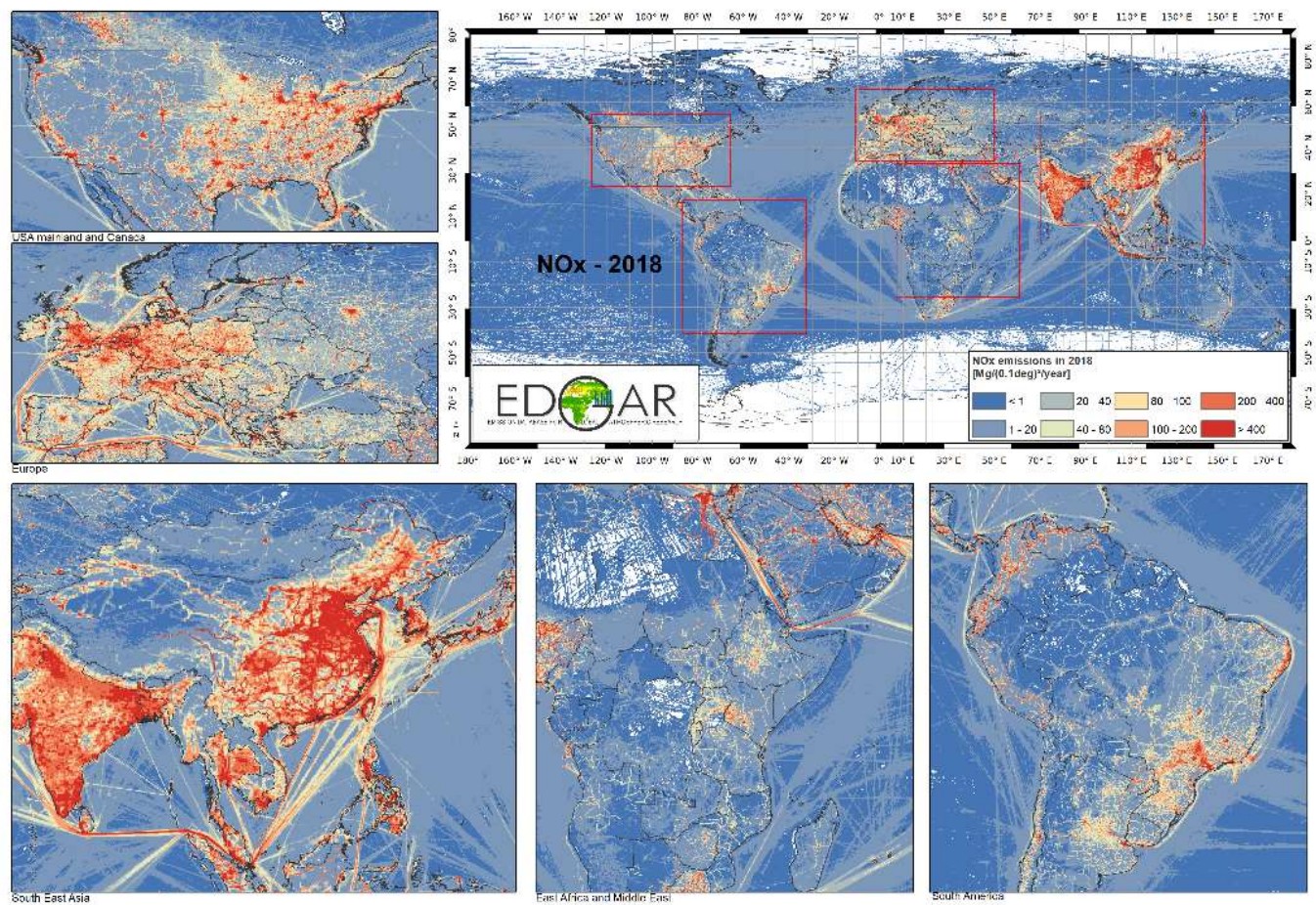

**Figure 5 – HTAP_v3 mosaic: NOx emission gridmaps in 2000 (top panel) and 2018 (bottom panel).**

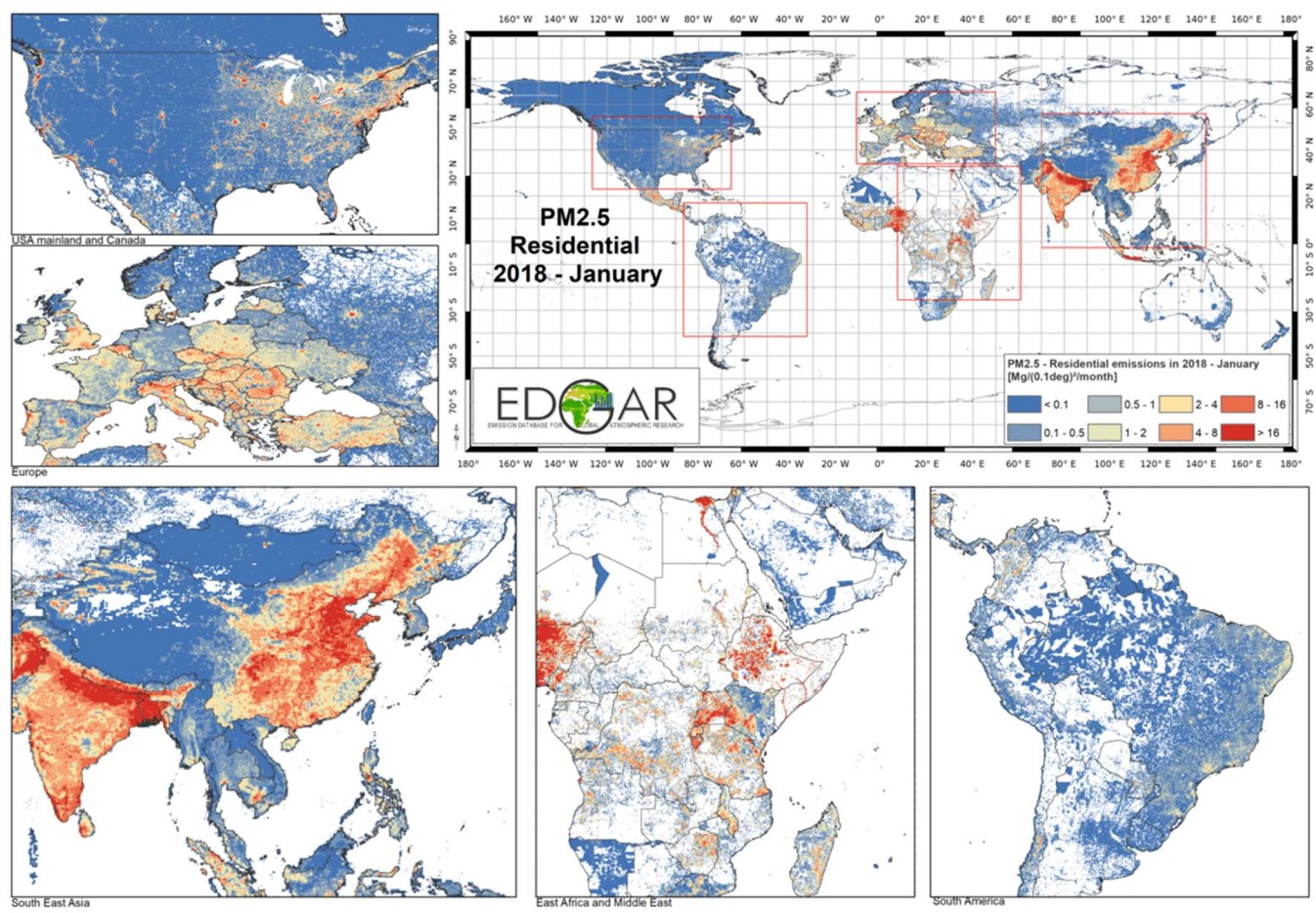

**Figure 6 – HTAP_v3 mosaic: PM$_{2.5}$ emissions from residential activities in January 2018.**

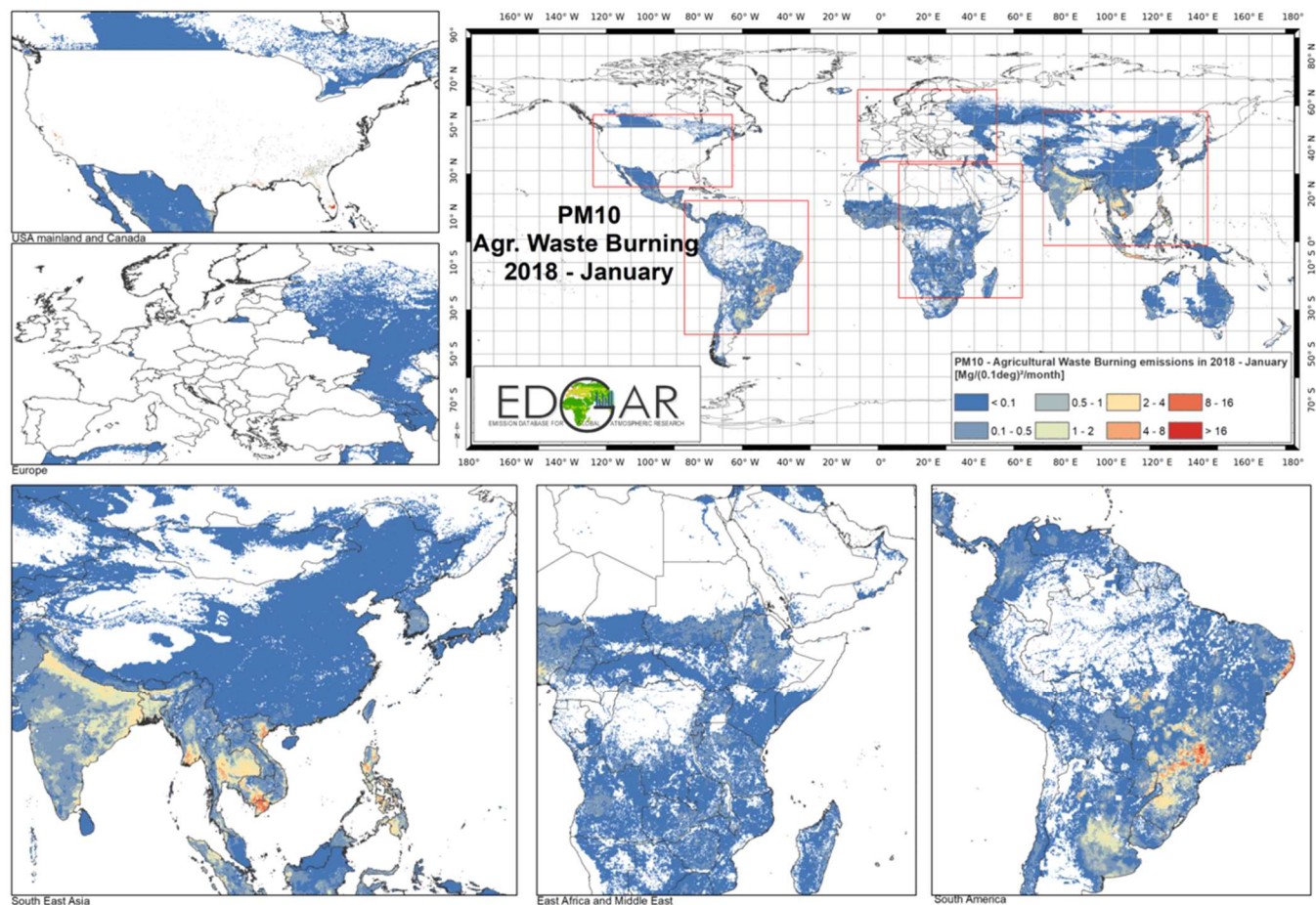

**Figure 7 – HTAP_v3 mosaic: PM$_{10}$ emissions from agricultural waste burning in January 2018.**

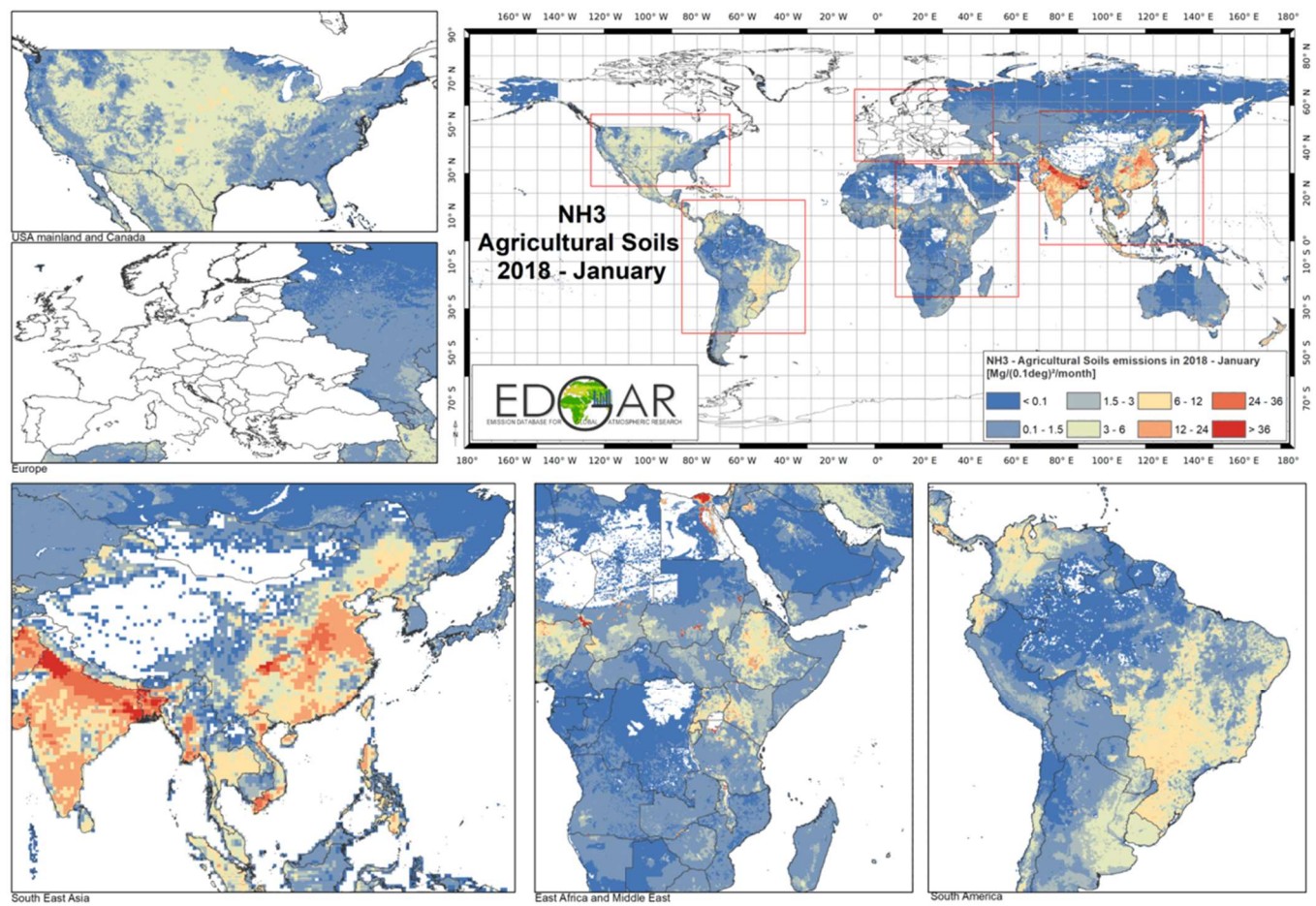

**Figure 8 – HTAP_v3 mosaic: NH₃ emissions from agricultural soils activities in January 2018.**

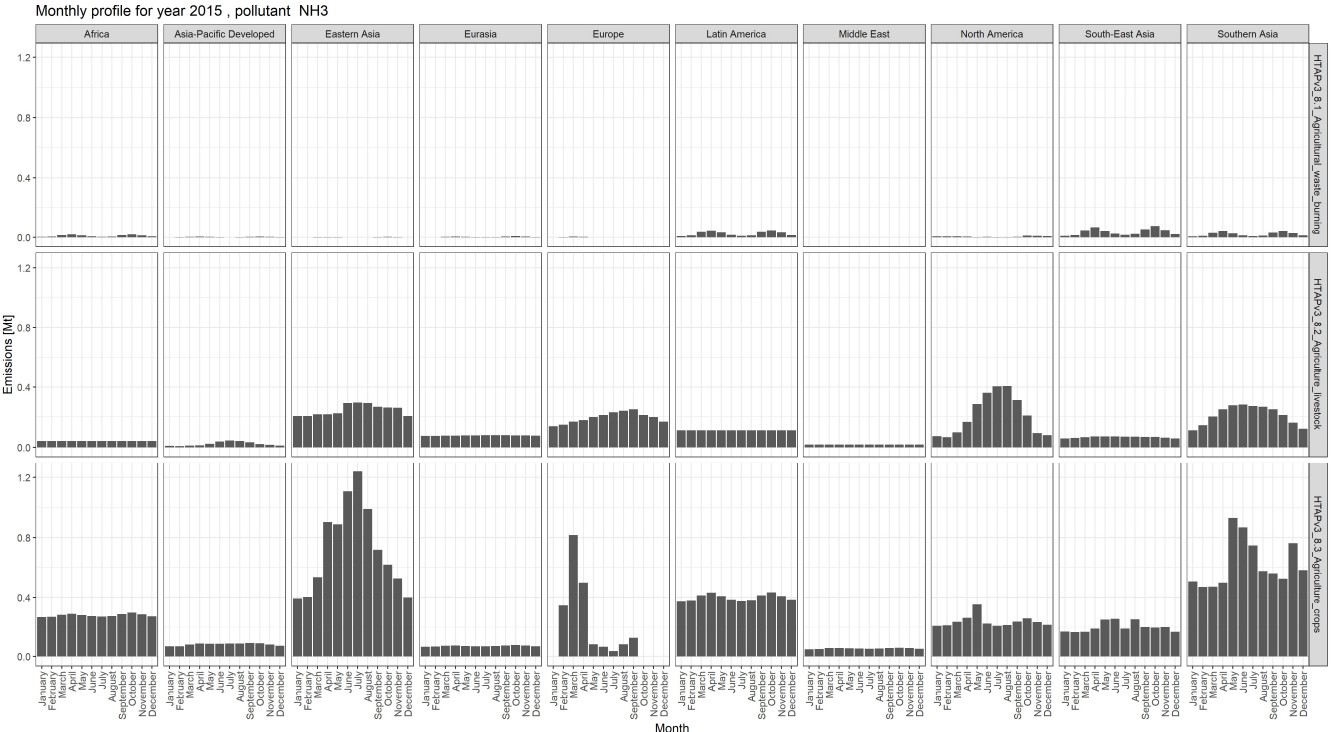

**Figure 9 – Monthly variability of NH₃ emissions for agriculture related activities for the different world regions in 2015.**

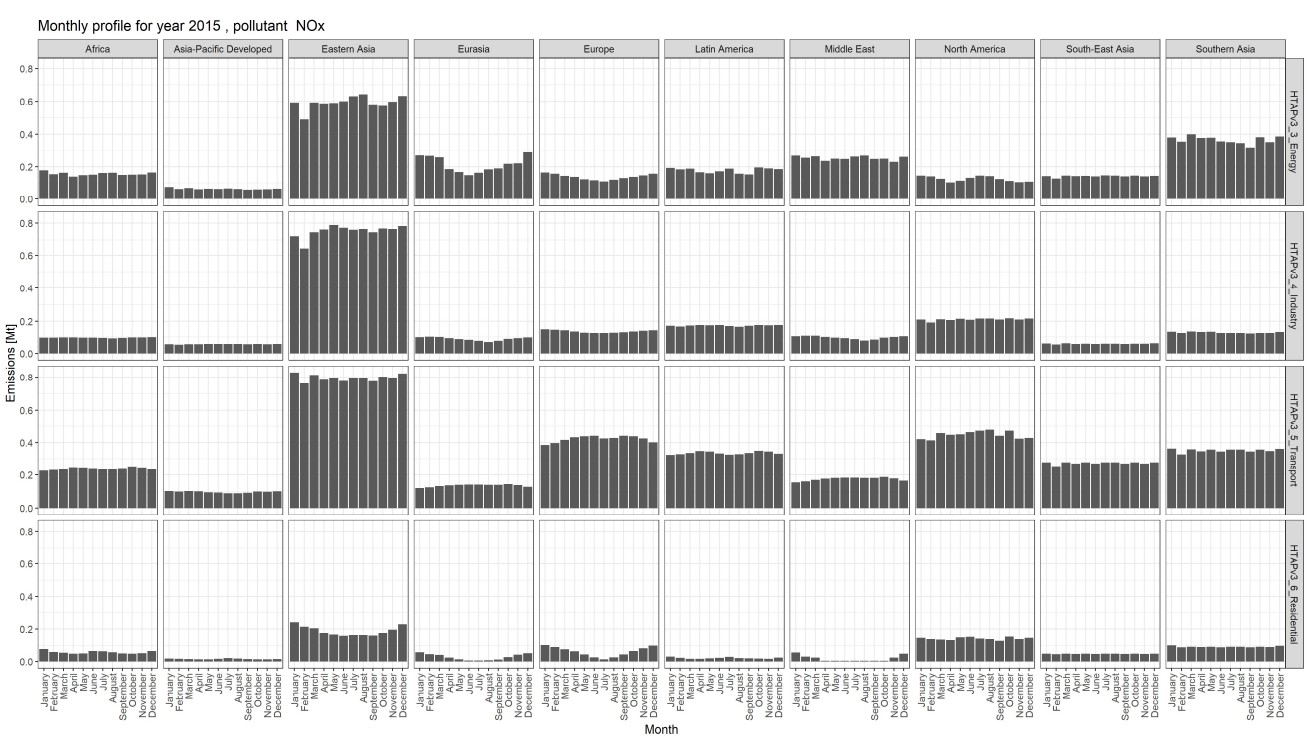

**Figure 10 – Monthly variability of NOx emissions for relevant emission sectors for the different world regions in 2015.**

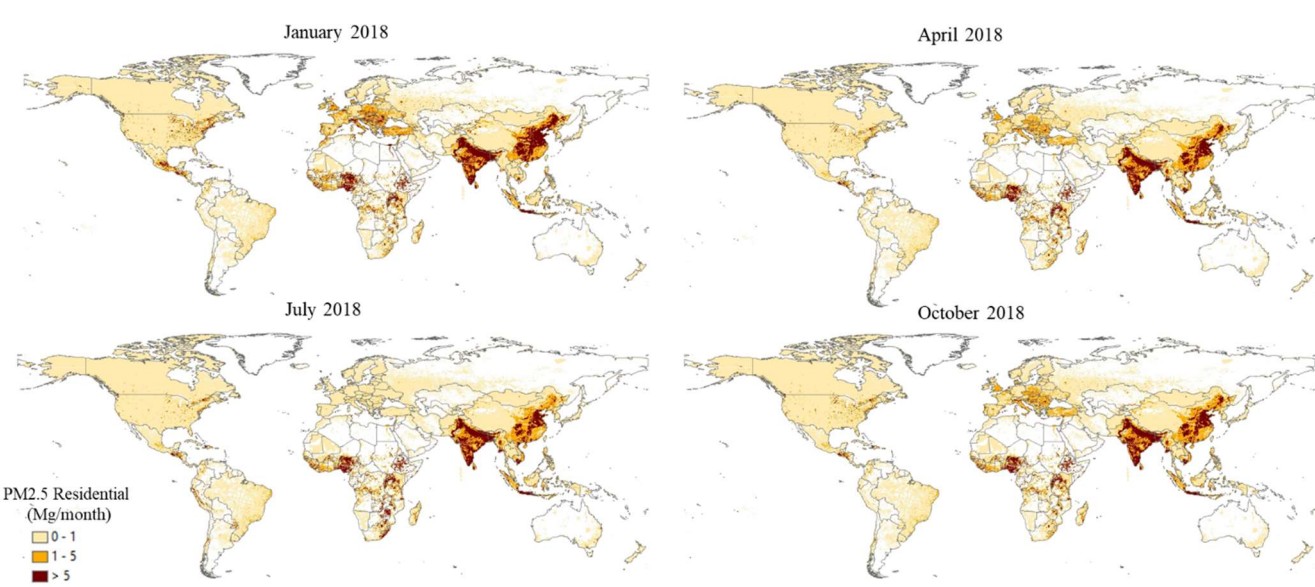

3  **Figure 11 – PM₂.₅ monthly emission maps from the residential sector in 2018 from**
4  **HTAP_v3.**

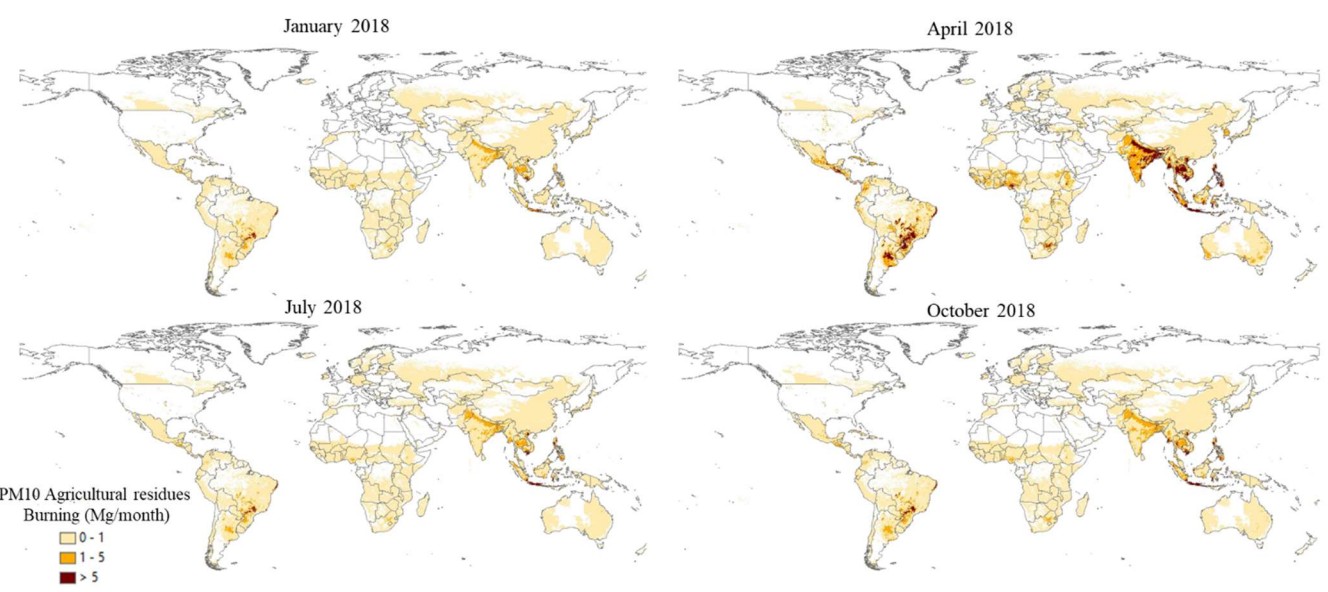

7  **Figure 12 – PM₁₀ monthly emission maps from agricultural residues burning in 2018 from**
8  **HTAP_v3.**

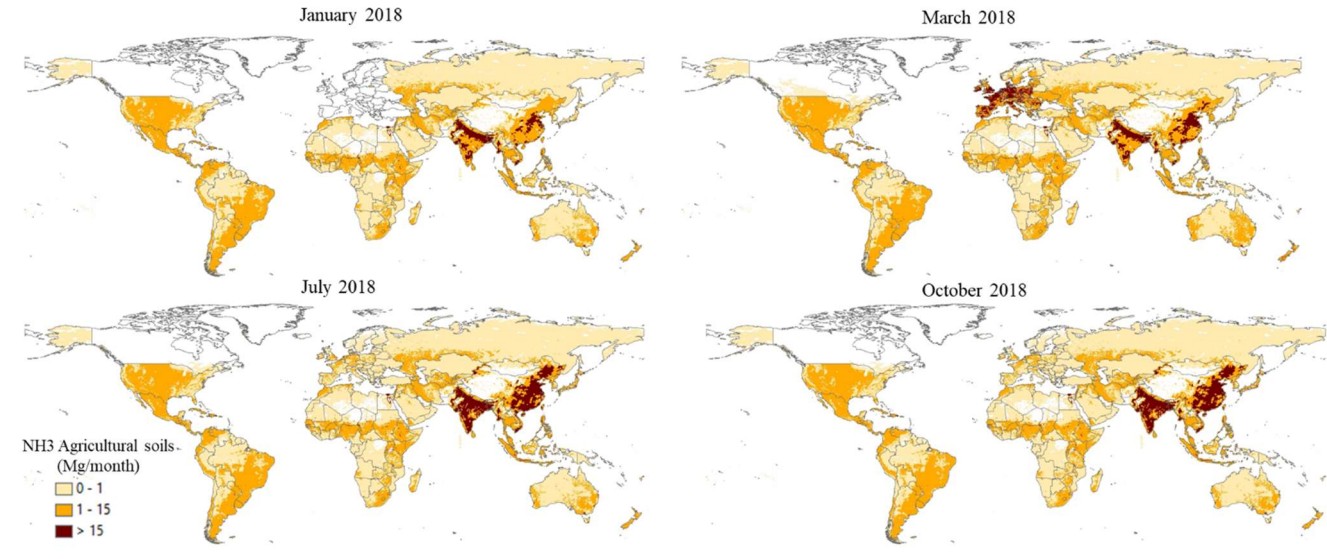

**Figure 13 – NH₃ monthly emission maps from agricultural soils in 2018 from HTAP_v3.**

**Table 1 – Overview of data input to the HTAP_v3 emission mosaic.**

| Data source | CAMS-REG-v5.1 | US EPA | ECCC | REASv3.2.1 |
|---|---|---|---|---|
| Type of data source | Country inventories as emission time series by sector and country and emission gridmaps as .csv files. | Country inventory | Country inventory as emission time series by sector and country and emission gridmaps as .NetCDF files. | Country inventories as emission gridmaps as text files. |
| Sectors coverage | All sectors, excluding international shipping and aviation (international and domestic). | All sectors, excluding international shipping and aviation (international and domestic). | All sectors, excluding agricultural waste burning, international shipping and aviation (international and domestic). | All sectors, excluding brake and tyre wear, domestic shipping, waste, agricultural waste burning, international shipping and aviation (international and domestic). |

| | | | | |
|---|---|---|---|---|
| **Temporal coverage** | 2000-2018 | 2002-2017 | 2000-2016 | 2000-2015+trends from MEIC over China for 2016, 2017, 2018 |
| **Temporal resolution** | Annual emission gridmaps + monthly profiles | Monthly emission gridmaps | Monthly emission gridmaps | Monthly emission gridmaps |
| **Spatial resolution** | 0.1°x0.1° | 0.1°x0.1° | 0.1°x0.1° | 0.1°x0.1° (The original spatial resolution of REASv3.2.1 is 0.25°x0.25°. Assuming that emissions are equally distributed in the 0.25° cell, REASv3.2.1 data were converted to 0.1° cell and provided to HTAP_v3) |
| **Substances** | SO2, NOx, CO, NMVOC, NH3, PM10, PM2.5, BC, OC | SO2, NOx, CO, NMVOC, NH3, PM10, PM2.5, BC, OC | SO2, NOx, CO, NMVOC, NH3, PM10, PM2.5, BC, OC | SO2, NOx, CO, NMVOC, NH3, PM10, PM2.5, BC, OC |
| **Geocoverage** |  |  |  |  |
| **References** | | | | http://meicmodel.org |

| Data source | CAPSS-KU | JAPAN (PM2.5EI and J-STREAM) | EDGARv6.1 | |
|---|---|---|---|---|
| **Type of data source** | Country inventory as emission time series by sector and country and emission gridmaps as .NetCDF files. | Country inventory as emission time series by sector and country and emission gridmaps as .NetCDF files. | Country inventory as emission time series by sector and country and emission gridmaps as .NetCDF files. | |
| **Sectors coverage** | All sectors, excluding international shipping and aviation (international and domestic). | All sectors, excluding international shipping, domestic shipping and aviation | All sectors, including international shipping and aviation (international and domestic) | |

| | | (international and domestic). | | |
|---|---|---|---|---|
| **Temporal coverage** | 2000-2018 | 2000-2017 | 2000-2018 | |
| **Temporal resolution** | Annual emission gridmaps + monthly profiles | Monthly emission gridmaps | Monthly emission gridmaps | |
| **Spatial resolution** | 0.1°x0.1° | 0.1°x0.1° | 0.1°x0.1° | |
| **Substances** | SO2, NOx, CO, NMVOC, NH3, PM10, PM2.5, BC, OC | SO2, NOx, CO, NMVOC, NH3, PM10, PM2.5, BC, OC | SO2, NOx, CO, NMVOC, NH3, PM10, PM2.5, BC, OC | |
| **Geocoverage** |  |  |  | |
| **References** | | | https://edgar.jrc.ec.europa.eu/dataset_ap61 | |

1    **Table 2 – Definition of HTAP_v3 sectors and correspondence to IPCC codes.**

| HTAP_v3 main sectors | HTAP_v3 detailed sectors | Sector description | IPCC 1996 codes | IPCC 2006 codes |
|---|---|---|---|---|
| HTAP_1: International Shipping | HTAP_1: International Shipping | International water-born navigation. | 1C2 | 1.A.3.d.i |
| HTAP_2: Aviation | HTAP_2.1: Domestic Aviation | Civil Aviation. | 1A3aii | 1.A.3.a.ii |
| | HTAP_2.2: International Aviation | International Aviation. | 1A3ai | 1.A.3.a.i |
| HTAP_3: Energy | HTAP_3: Energy | Power generation. | 1A1a | 1.A.1.a |
| HTAP_4: Industry | HTAP_4.1: Industry | Industrial non-power large-scale combustion emissions and emissions of industrial processes. It includes: manufacturing, mining, metal, cement, chemical and fossil fuel fires. | 1A2 + 2 + 5B | 1A2 + 2 (excluding 2.D.3 + 2.E + 2.F + 2.G) + 7A |
| | HTAP_4.2: Fugitive | It includes oil and gas exploration and production and transmission, including evaporative emissions (mainly NMVOC). | 1B + 1A1b + 1A1ci + 1A1cii + 1A5biii | 1.B + 1.A.1.b + 1.A.1.c.i + 1.A.1.c.i.i + 1.A.5.b.i.i.i |
| | HTAP_4.3: Solvents | Solvents and product use. | 3 | 2D3 + 2E + 2F + 2G |
| HTAP_5: Ground Transport | HTAP_5.1: Road Transport | Road Transport, combustion and evaporative emissions only. | 1A3b (excluding resuspension) | 1.A.3.b (excluding resuspension) |
| | HTAP_5.2: Brake and Tyre wear | Re-suspended dust from pavements or tyre and brake wear from road transport. | 1A3b (resuspension only) | 1.A.3.b (resuspension only) |
| | HTAP_5.3: Domestic shipping | Domestic shipping: inland waterways + domestic shipping. | 1A3d2 | 1.A.3.d.ii |
| | HTAP_5.4: Other ground transport | Ground transport by pipelines and other ground transport of mobile machinery. | 1A3c + 1A3e | 1.A.3.c + 1.A.3.e.ii |
| HTAP_6: Residential | HTAP_6: Residential | Small-scale combustion, including heating, cooling, lighting, cooking and auxiliary engines, to equip residential, | 1A4 + 1A5 | 1.A.4 + 1.A.5 |

| | | commercial buildings, service institutes, and agricultural facilities and fisheries. | | |
|---|---|---|---|---|
| **HTAP_7 : Waste** | **HTAP_7: Waste** | Solid waste disposal and wastewater treatment. | 6 | 4 |
| **HTAP_8 : Agricult ure** | **HTAP_8.1: Agricultural waste burning** | Agricultural waste burning (excluding Savannah burning). | 4F | 3.C.1.b |
| | **HTAP_8.2: Agriculture livestock** | Livestock emissions, including manure management. | 4B | 3.A.2 |
| | **HTAP_8.3: Agriculture crops** | Emissions from crops, fertilisers, and all agricultural soils activities. | 4C + 4D | 3.C.2 + 3.C.3 + 3.C.4 + 3.C.7 |

**Table 3 – Overview of pollutant and sector provided by each inventory in HTAP_v3. Cells**
**with N/A indicate that the emissions for those sectors were not provided and/or used in**
**HTAP_v3 for a specific inventory, while gapfilled with the corresponding information**
**from EDGARv6.1. The other cells represent the data availability for each sector and**
**inventory. The color codes used for the pollutants refer to the data source: black color**
**represents pollutant emissions provided by a specific inventory, red color indicates**
**emissions gapfilled using EDGARv6.1 and violet color indicates combinations of sectors-**
**pollutants available for specific regional inventories but not in EDGAR, which typically**
**represent minor sources of emissions included in officially reported inventories. These**
**minor sources are included in the HTAP_v3 mosaic.**

| **Data provider** | **REA Sv3.2 .1** | **CAP SS- KU** | **JAPAN** | **ECCC** | **US EPA** | **CAMS- REG- v5.1** | **EDGAR v6.1** |
|---|---|---|---|---|---|---|---|
| **HTAP_1: International Shipping** | N/A | N/A | N/A | N/A | N/A | N/A | All substanc es |
| **HTAP_2.1: Domestic Aviation** | N/A | N/A | N/A | N/A | N/A | N/A | All substanc es |
| **HTAP_2.2: International Aviation** | N/A | N/A | N/A | N/A | N/A | N/A | All substanc es |
| **HTAP_3: Energy** | All subst ances | All subst ances | BC, OC, NOx, NH3, CO, PM2.5, | All substanc es | All substanc es | All substanc es | All substanc es |

| | | | | | | | |
|---|---|---|---|---|---|---|---|
| | | | PM10, NMVOC, SO2 | | | | |
| **HTAP_4.1: Industry** | All substances | All substances | BC, OC, NOx, NH3, CO, PM2.5, PM10, NMVOC, SO2 | All substances | All substances | All substances | All substances |
| **HTAP_4.2: Fugitive** | All substances | BC, OC, NOx, NH3, CO, PM2.5, PM10, NMVOC, SO2 | BC, OC, NOx, NH3, CO, PM2.5, PM10, NMVOC, SO2 | All substances | All substances | All substances | All substances |
| **HTAP_4.3: Solvents** | NMVOC, NH3, PM10, PM2.5 | NMVOC, NH3, PM10, PM2.5 | NMVOC, NH3, PM10, PM2.5 | NMVOC, NH3, PM10, PM2.5 | CO, NOx, OC, NMVOC, NH3, PM10, PM2.5, SO2 | NOx, NH3, CO, PM2.5, PM10, NMVOC, SO2 | All substances |
| **HTAP_5.1: Road Transport** | All substances | All substances | All substances | All substances | All substances | All substances | All substances |
| **HTAP_5.2: Brake and Tyre wear** | N/A | BC, OC, NOx, NH3, CO, PM2.5, PM10, NMVOC, SO2 | BC,OC, PM2.5,PM10 | BC,OC, PM2.5,PM10 | BC,OC, PM2.5,PM10, NMVOC | BC,OC, PM2.5,PM10 | BC,OC, PM2.5,PM10 |
| **HTAP_5.3: Domestic shipping** | N/A | All substances | N/A | All substances | All substances | BC, OC, NOx, NH3, CO, | All substances |

| | | | | | | | |
|---|---|---|---|---|---|---|---|
| | | | | | | PM2.5, PM10, NMVOC, SO2 | |
| **HTAP_5.4: Other ground transport** | All substances | All substances | BC, OC, NOx, NH3, CO, PM2.5, PM10, NMVOC, SO2 | All substances | All substances | All substances | All substances |
| **HTAP_6: Residential** | All substances | All substances | BC, OC, NOx, NH3, CO, PM2.5, PM10, NMVOC, SO2 | All substances | All substances | All substances | All substances |
| **HTAP_7: Waste** | **N/A** | All substances | All substances | All substances | All substances | All substances | All substances |
| **HTAP_8.1: Agricultural waste burning** | **N/A** | All substances | All substances | **N/A** | All substances | All substances | All substances |
| **HTAP_8.2: Agriculture livestock** | NH3, NMVOC, NOx, PM10, PM2.5 | NH3, NMVOC, NOx, PM10, PM2.5, BC, OC | NH3, NMVOC, NOx, PM10, PM2.5 | NH3, NMVOC, NOx, PM10, PM2.5, BC, OC | NH3, NMVOC, NOx, PM10, PM2.5, BC, OC | NH3, NMVOC, NOx, PM10, PM2.5, OC | NH3, NMVOC, NOx, PM10, PM2.5 |
| **HTAP_8.3: Agriculture_crops** | NH3, NOx, PM10, PM2.5 | NH3, NOx, PM10, PM2.5, BC, OC | NH3, NOx, PM10, PM2.5 | NH3, NOx, PM10, PM2.5, BC, NMVOC, OC | NH3, NOx, PM10, PM2.5, BC, OC, CO, NMVOC, SO2 | NH3, NOx, PM10, PM2.5, SO2, CO, OC, NMVOC | NH3, NOx, PM10, PM2.5 |

Table 4 – Main features of the different HTAP mosaics.

|  | HTAP_v1 | HTAP_v2.2 | HTAP_v3 |
|---|---|---|---|
| **Time coverage** | 2000-2005 | 2008 and 2010 | 2000-2018 |
| **Time resolution** | yearly | yearly and monthly | yearly and monhtly |
| **Substances** | $CH_4$, NMVOC, CO, $SO_2$, NOx, $NH_3$, $PM_{10}$, $PM_{2.5}$, BC, OC | $SO_2$, NOx, CO, NMVOC, $NH_3$ (only for agriculture), $PM_{10}$, $PM_{2.5}$, BC, OC | $SO_2$, NOx, CO, NMVOC, $NH_3$, $PM_{10}$, $PM_{2.5}$, BC, OC |
| **Sectors** | Aircraft, Ships, Energy, Industry Processes, Ground Transport, Residential, Solvents, Agriculture, Agriculture Waste Burning, and Waste | Air, Ships, Energy, Industry, Transport, Residential (including waste), and Agriculture (only for NH3) | International Shipping, Domestic Shipping, Domestic Aviation, International Aviation, Energy, Industry, Fugitives, Solvent Use, Road Transport, Brake and Tyre Wear, Other Ground Transport, Residential, Waste, Agricultural Waste Burning, Livestock, and Agricultural Crops |
| **Geographical coverage** | Globe | Globe | Globe |
| **Spatial resolution** | 0.1°x0.1° | 0.1°x0.1° | 0.1°x0.1° |
| **Input datasets** | UNFCCC, REAS, GAINS, EMEP, EPA, EDGARv4.1 | US EPA, Environment Canada, MICS, TNO/EMEP Europe (MACC II), MICS Asia III+ REAS2.1, EDGARv4.3 | CAMS-REG-v5.1, REASv3.2.1, US EPA, ECCC, CAPSS-KU, JAPAN (PM2.5EI and J-STREAM), EDGARv6.1 |
| **Reference** | Janssens-Maenhout et al., 2012 | Janssens-Maenhout et al., 2015 | This work |

