# Peer review of "The HTAP_v3 emission mosaic: merging regional and global monthly emissions (2000-2018) to support air quality modelling and policies"

_Earth System Science Data, 2022_

## Author Comment (AC1)

The authors are grateful to both reviewers for the constructive comments that helped to improve the quality of the manuscript. Please see below our answers (in red) to the Reviewers' comments.

**Anonymous Referee #1**

This manuscript presents a description of the HTAPv3 global mosaic of anthropogenic inventories. The dataset provides a consistent times series for almost two decades of air pollutant emissions at high spatial (0.1x0.1 degree) and temporal (monthly) resolution by incorporating the best available local information. As stated by the authors, HTAPv3 is a unique and state-of-the-art tool that will substantially contribute to support policy-relevant modelling studies at both regional and global levels. Therefore, the dataset presented in the manuscript if of interest. The paper is well written and structured, and its quality is very good, which makes it a good contribution to ESSD. I recommend the manuscript to be published once the following comments have been addressed:

**General comments:**
Despite the recent advances made in terms of emission inventory developments in Latin America and Africa - e.g., GEAA-AEIv3.0M for Argentina by Puliafito et al. (2021), INEMA for Chile by Álamos et al., (2022); DACCIWA for Africa by Keita et al., (2021) – the HTAP_v3 mosaic does not integrate regional inventories from any of these two regions. Could you clarify why you decided to cover these regions with EDGAR emissions instead of using more local information?

As correctly pointed out by the Reviewer, the HTAP_v3 mosaic aims to represent historic and current emission levels distributed over the globe, integrating to the extent possible officially reported emission maps and state-of-the-art inventories. When the third phase of the HTAP_v3 emission mosaic was launched (Spring 2020) a consultation involving all emission inventory developers and the broader scientific community was conducted with the aim of identifying data providers for each region and defining technical details (e.g., sector disaggregation, time frame, data formats, etc.) as well as the willingness to commit to extensive cooperation with the mosaic development team in order to ensure that each regional inventory was appropriately integrated into the mosaic. The opening of the HTAP_v3 mosaic data collection task was also announced at the TF-HTAP meeting in April 2020, and every year since a follow-up presentation at the Spring TF-HTAP meeting has been performed in order to communicate to a larger community the ongoing work, to ask for feedback, and to expand participation by the broader community.

The HTAP_v3 mosaic does not represent only a collection of publicly available data, but also the community effort made by researchers and experts in the field of emissions working together to enhance our understanding of global air pollutant emission sources. In all cases, extensive cooperation between the mosaic development team and each regional team was required to ensure compatibility, identify sectoral gaps, and avoid double-counting of emissions.

After assessing the interest shown by different inventory compilers, the corresponding data availability, and the time schedule foreseen for this project (publication of the HTAP_v3 mosaic by early 2022), the current data providers of HTAP_v3 (CAMS-REG, REAS, USA EPA, ECCC, CAPSS-KU, JAPAN (PM2.5EI and J-STREAM), EDGAR) were selected. Furthermore, looking at the hemispheric transport of air pollution a major role is played by

regions situated in the Northern Hemisphere which also reinforced our decision to start working with people already available for collaboration and then gap-filling missing information with the EDGAR database.

While the design and development of the HTAP_v3 mosaic required focussed time and effort over a short time period, looking now at newly available datasets such as the GEAA-AEIv3.0M for Argentina by Puliafito et al. (2021), the INEMA for Chile by Álamos et al., (2022), and the DACCIWA for Africa by Keita et al., (2021) we believe that including some of these inventories in an updated version of HTAP_v3 would be relevant. However, the modelling community needs the HTAP_v3 data now and their modelling experiments cannot be delayed.

More specifically, the inventory for Argentina documented by Puliafito et al. (2021) seems to fulfil all of the requirements of HTAP_v3 (e.g. temporal, spatial, and sectoral coverage), and the authors believe it would represent an important improvement to characterize South American emissions, although covering only Argentina.

The inventory for Chile (Alamos et al., 2022), on the other hand, covers only the 2015-2017 period, while the purpose of HTAP_v3 was to support time series analysis of air pollutant emissions and the corresponding air quality modelling. So, the work by Alamos et al. (2022), although possibly representing the best available knowledge of recent emission levels in Chile, does not satisfy the selection criteria defined for the development of HTAP_v3.

Concerning the work by Huneeus et al. (2020), it provides an overview of available global inventories (including EDGAR) and local/city scale datasets. This work aims at evaluating differences among inventories and approaches, but it does not provide a reference official inventory to be used over the South American domain. For this reason, only the Puliafito et al. (2021) work covering Argentina may be considered in future updates of HTAP_v3. Another possibility would be to rely on the ongoing work under the 'Latin America and Caribbean GEIA Working Group' (http://www.geiacenter.org/analysis/working-groups/latin-america-and-caribbean-wg) and integrate state-of-the-art inventories for South American countries as soon as they become available. This process will require time and coordination with local experts, therefore being an ongoing work it cannot be integrated in the current HTAP_v3 mosaic.

The DACCIWA inventory for Africa (Keita et al., 2021) was not yet published at the time when we gathered the contributions from the different data providers. Feeding the HTAP_v3 mosaic with the DACCIWA data would require the support of the DACCIWA developers since higher sector resolution data should be shared, consistent with HTAP_v3 features. Furthermore, the DACCIWA data require authorization for download.

To summarize, we agree with the suggestion of the Reviewer regarding the possibility to include inventories covering the South American and African domains in future updates of HTAP_v3 in collaboration with the corresponding inventory developers.

In the conclusions we added the following paragraph to possibly include in future updates of HTAP_v3 additional regional inventories following the scientific literature development:

'Thanks to the continuous improvement of local and regional emission inventories, recent literature shows emerging new datasets reporting regional information over areas of the world not covered by local inventories in the current HTAP_v3 mosaic (e.g. Argentina (Puliafito et

al. 2021), Africa (Keita et al., 2021) or the MEIC inventory (http://meicmodel.org.cn/?page_id=1772&lang=en)). Future updates of this work may also integrate reliable and up to data information over South America or Africa as time and resources permit.'

Following with the previous point, it is not clear to me why for China the authors decided to use REAS instead of MEIC, giving the fact that the later report emissions until a more recent year (2015 versus 2017) and that the information considered to spatially distribute emissions from industrial plants is more precise in MEIC.

Similarly to the arguments reported above in the selection of the inventories contributing to HTAP_v3, in the case of MEIC the authors were not able to obtain detailed data (e.g., sector disaggregation, temporal coverage, and high spatial resolution maps) and all of the support from the MEIC experts needed to appropriately integrate the MEIC data for China into the HTAP_v3 mosaic. MEIC publicly available data (http://meicmodel.org.cn/?page_id=1772&lang=en) unfortunately do not follow the requirements of the HTAP_v3 mosaic, so enhancing cooperation with the MEIC developers is of the utmost importance in future when representing Chinese emissions. We hope to be able to include Chinese emissions from MEIC in a future HTAP mosaic inventory through the necessary cooperation with the MEIC experts.

According to the authors, "One key goal of the HTAP_v3 mosaic is to collate in one inventory the most accurate spatially-distributed emissions for all air pollutants at the global level, based on the best available local information". Recent studies have shown that the EDGAR inventory tends to significantly over allocate PM emissions from residential combustion processes in certain urban areas of Latin America (Huneeus et al., 2020). Coming back to my first point, should not HTAPv3 consider local available emissions for this region?

In line with the comments of the Reviewer regarding specific local inventories which could have been incorporated in HTAP_v3, the following paragraph has been included in section 2.1 to clarify the choice of the selected local/regional inventories and possible future updates of HTAP_v3:

'Recent literature studies (Puliafito et al., 2021; Huneeus et al., 2020; Alamos et al., 2022; Keita et al., 2021; MEIC for China (http://meicmodel.org.cn/?page_id=1772&lang=en)) document additional regional/local inventories which may contribute to future updates of HTAP_v3, in particular extending the mosaic compilation to regions in the Southern Hemisphere. Considering relative hemispheric emission levels as well as the atmospheric dynamics happening in the Northern Hemisphere and regulating the transboundary transport of air pollution, the current HTAP_v3 mosaic should still satisfy the needs of the atmospheric modelling community, although improvements using latest available inventories for Africa and South America may also be considered for future updates.'

The HTAPv3 inventory includes NOx and NMVOC emissions from agricultural crops. These emissions could potentially be double counted if HTAPv3 is combined with a natural emission model such as MEGAN, which includes the estimation of NMVOC from crops and soil NOx emissions (including agricultural soils). It would be good if the authors can add a sentence mentioning that these emissions should be treated with careful.

The authors agree with the Reviewer's comment and the following paragraph is now included in the paper:

'The high sector disaggregation available within the HTAP_v3 mosaic gives needed flexibility to modellers to include or exclude emission sub-sectors in their simulations, in particular when integrating the anthropogenic emissions provided by HTAP_v3 with other components (e.g. natural emissions, forest fires, etc.). However, we recommend particular caution when using a natural emissions model such as MEGAN (Model of Emissions of Gases and Aerosols from Nature, https://www2.acom.ucar.edu/modeling/model-emissions-gases-and-aerosols-nature-megan), which includes the estimation of NMVOC emissions from crops and soil NOx emissions (including agricultural soils) that are also provided by the HTAP_v3 mosaic.'

The HTAPv3 inventory provides information and guidance for the speciation of NMVOC and PM emissions (PM2.5 is reported together with BC and OC). However, no information is provided concerning the speciation of NOx emissions (NO and NO2, and HONO for the specific case of road transport). Could you comment on this point?

In HTAP_v3, NOx emissions include the sum of NO+NO2 and they are expressed as NO2 mass unit. In HTAP_v3, NOx emissions include the sum of NO+NO2 and they are expressed as NO2 mass unit. For road transport, we expect the partitioning of NOx emissions between NO, NO2, and HONO to be highly region-dependent based on the fleet composition (e.g., number of diesel vehicles relative to gasoline vehicles) and technology level (e.g., the level of exhaust after treatment). The regional inventories, however, report total NOx with no speciation. Standard practice in global models is to emit all anthropogenic NOx as NO, while we expect that regional modelling groups will have access to appropriate best practices for their particular regions. This was our experience with previous HTAP mosaic inventories. We therefore consider a speciation for NOx emissions in the HTAPv3 global mosaic to be beyond the scope of the current manuscript. If the community expresses a strong interest in speciated NOx emissions, we may revisit this for future versions of the HTAP mosaic inventories. This is now clarified in the revised manuscript (conclusions section) as following:

'The speciation of NOx emissions into its components (NO, NO2, HONO) is not provided by the global HTAP_v3 mosaic and it is beyond the scope of the current work since the regional inventories report total NOx with no speciation. Standard practice in global models is to emit all anthropogenic NOx as NO, while we expect that regional modelling groups will have access to appropriate best practices for their particular regions. In particular for road transport, the partitioning of NOx emissions between NO, NO2, and HONO is highly region-dependent and it is based on the fleet composition (e.g., number of diesel vehicles relative to gasoline vehicles) and technology level (e.g., the level of exhaust after treatment).'

Following with what happened with their predecessors, HTAP_v3 will quickly become a widely used emission dataset among the air quality modelling community. Having this in mind, I think it is important that in the conclusions sections the authors include a subsection listing the main limitations of the dataset and summarizing the considerations that users should take into account when using it (e.g., agricultural waste burning emissions should be treated with caution to avoid double counting when combined with existing biomass burning emission inventories). I would also recommend to further develop the part on future works (e.g., are there any knowing emission information needs from the modelling community that could not be covered with the present version of HTAP and may be tackled in future versions?)

We agreed with the Reviewer's comment and therefore we have added a section to the conclusions discussing the main limitations and future updates of HTAP_v3:

'When using the HTAP_v3 emission mosaic, users should consider the following limitations, for example when combining the HTAP_v3 data with other emission input needed to run atmospheric models:

- agricultural waste burning emissions should be treated with caution to avoid double-counting when combined with existing biomass burning emission inventories;

- NMVOC and NOx emissions from agricultural soils should be treated with caution to avoid double-counting when combining the HTAP_v3 data with a natural emissions model such as MEGAN (Model of Emissions of Gases and Aerosols from Nature);

- the speciation of NOx emissions into its components (NO, NO2, HONO) is not provided by the global HTAP_v3 mosaic and it is beyond the scope of the current work since the regional inventories report total NOx with no speciation. Standard practice in global models is to emit all anthropogenic NOx as NO, while we expect that regional modelling groups will have access to appropriate best practices for their particular regions. In particular for road transport, the partitioning of NOx emissions between NO, NO2, and HONO is highly region-dependent and it is based on the fleet composition (e.g., number of diesel vehicles relative to gasoline vehicles) and technology level (e.g., the level of exhaust after treatment).

Thanks to the continuous improvement of local and regional emission inventories, recent literature shows new datasets that report regional information over areas of the world not covered by local inventories in the current HTAP_v3 mosaic (e.g. Argentina (Puliafito et al. 2021), Africa (Keita et al., 2021) and the MEIC inventory (http://meicmodel.org.cn/?page_id=1772&lang=en)). Future updates to this mosaic may also integrate reliable and up to data information over South America or Africa as time and resources permit.'

**Specific comments:**

Adding a summary table that compares the main features of the different versions of HTAP (e.g., pollutants, number of sectors, temporal coverage, resolution), would help the reader to quickly spot the main improvements and added value of this new version Change CAMS-REF-v5.1 to CAMS-REG-v5.1 (line 54)

The correction of CAMS-REG-v5.1 has been included.

A summary table including the comparison of the main features of the three HTAP mosaics has been includes as Table 4, as shown below:

| | HTAP_v1 | HTAP_v2.2 | HTAP_v3 |
|---|---|---|---|
| **Time coverage** | 2000-2005 | 2008 and 2010 | 2000-2018 |
| **Time resolution** | yearly | yearly and monthly | yearly and monhtly |
| **Substances** | $CH_4$, NMVOC, CO, $SO_2$, NOx, $NH_3$, $PM_{10}$, $PM_{2.5}$, BC, OC | $SO_2$, NOx, CO, NMVOC, $NH_3$ (only for agriculture), $PM_{10}$, $PM_{2.5}$, BC, OC | $SO_2$, NOx, CO, NMVOC, $NH_3$, |

| | | | |
|---|---|---|---|
| | | | PM$_{10}$, PM$_{2.5}$, BC, OC |
| **Sectors** | Aircraft, Ships, Energy, Industry Processes, Ground Transport, Residential, Solvents, Agriculture, Agriculture Waste Burning, and Waste | Air, Ships, Energy, Industry, Transport, Residential (including waste), and Agriculture (only for NH3) | International Shipping, Domestic Shipping, Domestic Aviation, International Aviation, Energy, Industry, Fugitives, Solvent Use, Road Transport, Brake and Tyre Wear, Other Ground Transport, Residential, Waste, Agricultural Waste Burning, Livestock, and Agricultural Crops |
| **Geographical coverage** | Globe | Globe | Globe |
| **Spatial resolution** | 0.1°x0.1° | 0.1°x0.1° | 0.1°x0.1° |
| **Input datasets** | UNFCCC, REAS, GAINS, EMEP, EPA, EDGARv4.1 | US EPA, Environment Canada, MICS, TNO/EMEP Europe (MACC II), MICS Asia III+ REAS2.1, EDGARv4.3 | CAMS-REG-v5.1, REASv3.2.1, US EPA, ECCC, CAPSS-KU, JAPAN (PM2.5EI and J-STREAM), EDGARv6.1 |
| **Reference** | Janssens-Maenhout et al., 2012 | Janssens-Maenhout et al., 2015 | This work |

**References:**

Puliafito, S. E., Bolaño-Ortiz, T. R., Fernandez, R. P., Berná, L. L., Pascual-Flores, R. M., Urquiza, J., López-Noreña, A. I., and Tames, M. F.: High-resolution seasonal and decadal inventory of anthropogenic gas-phase and particle emissions for Argentina, Earth Syst. Sci. Data, 13, 5027–5069, https://doi.org/10.5194/essd-13-5027-2021, 2021.

Keita, S., Liousse, C., Assamoi, E.-M., Doumbia, T., N'Datchoh, E. T., Gnamien, S., Elguindi, N., Granier, C., and Yoboué, V.: African anthropogenic emissions inventory for gases and particles from 1990 to 2015, Earth Syst. Sci. Data, 13, 3691–3705, https://doi.org/10.5194/essd-13-3691-2021, 2021.

Álamos, N., Huneeus, N., Opazo, M., Osses, M., Puja, S., Pantoja, N., Denier van der Gon, H., Schueftan, A., Reyes, R., and Calvo, R.: High-resolution inventory of atmospheric emissions from transport, industrial, energy, mining and residential activities in Chile, Earth Syst. Sci. Data, 14, 361–379, https://doi.org/10.5194/essd-14-361-2022, 2022.

Nicolas Huneeus, Hugo Denier van Der Gon, Paula Castesana, Camilo Menares, Claire Granier, et al.. Evaluation of anthropogenic air pollutant emission inventories for South America at national and city scale. Atmospheric Environment, 2020, 235, pp.117606. 10.1016/j.atmosenv.2020.117606.

---

## Author Comment (AC2)

The authors are grateful to both reviewers for the constructive comments that helped to improve the quality of the manuscript. Please see below our answers (in red) to the Reviewers' comments.

**Review 2 (Hugo Denier van der Gon)**

The paper describes the construction of an emission database using where possible national or regional inventories and using a global dataset (EDGARv6) for gap filling and achieving completeness. A major difference with previous efforts (e.g. HTAPv2) is that a complete timeseries is provided and that much more recent years are covered. The dataset meets all the requirement of (global) AQ modellers and at the same time is highly policy relevant because generally speaking the regional inventories are closest to nationally-accepted emission levels. Moreover the authors made a great effort to accommodate potential users for example with sector-cross walk tables, tools to extract certain domains and a consisting mapping of VOC species. The paper is a valuable contribution to the scientific community and deserves to be published after a number of, mostly minor, corrections are made and a few issues are discussed in a bit more detail.

I only select major revision because I really like to EDGARv6 also in the figures S1-S4. (see the comments below)

The EDGARv6 emissions are now included in Figures S1-S5.

**Main concerns**

Title: The title does not make clear that regional and global emission datasets are merged nor that it is a timeseries . I suggest changing the title. A suggestion could be (but not necessarily the best)

*"The HTAP_v3 emission mosaic: merging regional and global emission time series (2000-2018) to support air quality modelling and policies"*

We revised the title including some suggestions of the Reviewer as following:

*"The HTAP_v3 emission mosaic: merging regional and global monthly emissions (2000-2018) to support air quality modelling and policies"*

A more complete discussion on the use of REAS vs MEIC for China should be provided. In the Supp material (page 13) it is stated that *"CEDS is calibrated to MEIC, as is HTAP_v3 (indirectly via REAS)"*. So are the MEIC emissions the same as REAS? In the paper this is not clear, as it says REAS is scaled to MEIC for the years after 2015. Scaling is more about trends, calibration is more about absolute levels. Since China is globally such a major contributor it would be good to be more explicit about why MEIC is not used directly (there can be very valid reasons) and what is meant with indirect calibration and what the scaling does.

Regarding the reasons for not directly including the MEIC inventory, we refer the reviewer to our responses to anonymous referee #1 on the same question (as reported here below). Regarding the scaling of REAS emissions after 2015, we can confirm that the scaling was done

with the trends from MEIC, and not the absolute emissions. The sentence of the Supplement mentioned by the Reviewer has been revised accordingly.

*As correctly pointed out by the Reviewer, the HTAP_v3 mosaic aims to represent historic and current emission levels distributed over the globe, integrating to the extent possible officially reported emission maps and state-of-the-art inventories. When the third phase of the HTAP_v3 emission mosaic was launched (Spring 2020) a consultation involving all emission inventory developers and the broader scientific community was conducted with the aim of identifying data providers for each region and defining technical details (e.g., sector disaggregation, time frame, data formats, etc.) as well as the willingness to commit to extensive cooperation with the mosaic development team in order to ensure that each regional inventory was appropriately integrated into the mosaic. The opening of the HTAP_v3 mosaic data collection task was also announced at the TF-HTAP meeting in April 2020, and every year since a follow-up presentation at the Spring TF-HTAP meeting has been performed in order to communicate to a larger community the ongoing work, to ask for feedback, and to expand participation by the broader community.*

*The HTAP_v3 mosaic does not represent only a collection of publicly available data, but also the community effort made by researchers and experts in the field of emissions working together to enhance our understanding of global air pollutant emission sources. In all cases, extensive cooperation between the mosaic development team and each regional team was required to ensure compatibility, identify sectoral gaps, and avoid double-counting of emissions.*

*After assessing the interest shown by different inventory compilers, the corresponding data availability, and the time schedule foreseen for this project (publication of the HTAP_v3 mosaic by early 2022), the current data providers of HTAP_v3 (CAMS-REG, REAS, USA EPA, ECCC, CAPSS-KU, JAPAN (PM2.5EI and J-STREAM), EDGAR) were selected. Furthermore, looking at the hemispheric transport of air pollution a major role is played by regions situated in the Northern Hemisphere which also reinforced our decision to start working with people already available for collaboration and then gap-filling missing information with the EDGAR database.*

*While the design and development of the HTAP_v3 mosaic required focussed time and effort over a short time period, looking now at newly available datasets such as the GEAA-AEIv3.0M for Argentina by Puliafito et al. (2021), the INEMA for Chile by Álamos et al., (2022), and the DACCIWA for Africa by Keita et al., (2021) we believe that including some of these inventories in an updated version of HTAP_v3 would be relevant. However, the modelling community needs the HTAP_v3 data now and their modelling experiments cannot be delayed.*

*More specifically, the inventory for Argentina documented by Puliafito et al. (2021) seems to fulfil all of the requirements of HTAP_v3 (e.g. temporal, spatial, and sectoral coverage), and the authors believe it would represent an important improvement to characterize South American emissions, although covering only Argentina.*

*The inventory for Chile (Alamos et al., 2022), on the other hand, covers only the 2015-2017 period, while the purpose of HTAP_v3 was to support time series analysis of air pollutant emissions and the corresponding air quality modelling. So, the work by Alamos et al. (2022), although possibly representing the best available knowledge of recent emission levels in Chile, does not satisfy the selection criteria defined for the development of HTAP_v3.*

*Concerning the work by Huneeus et al. (2020), it provides an overview of available global inventories (including EDGAR) and local/city scale datasets. This work aims at evaluating differences among inventories and approaches, but it does not provide a reference official inventory to be used over the South American domain. For this reason, only the Puliafito et al. (2021) work covering Argentina may be considered in future updates of HTAP_v3. Another possibility would be to rely on the ongoing work under the 'Latin America and Caribbean GEIA Working Group' (http://www.geiacenter.org/analysis/working-groups/latin-america-and-caribbean-wg) and integrate state-of-the-art inventories for South American countries as soon as they become available. This process will require time and coordination with local experts, therefore being an ongoing work it cannot be integrated in the current HTAP_v3 mosaic.*

*The DACCIWA inventory for Africa (Keita et al., 2021) was not yet published at the time when we gathered the contributions from the different data providers. Feeding the HTAP_v3 mosaic with the DACCIWA data would require the support of the DACCIWA developers since higher sector resolution data should be shared, consistent with HTAP_v3 features. Furthermore, the DACCIWA data require authorization for download.*

*To summarize, we agree with the suggestion of the Reviewer regarding the possibility to include inventories covering the South American and African domains in future updates of HTAP_v3 in collaboration with the corresponding inventory developers.*

*In the conclusions we added the following paragraph to possibly include in future updates of HTAP_v3 additional regional inventories following the scientific literature development:*

*'Thanks to the continuous improvement of local and regional emission inventories, recent literature shows emerging new datasets reporting regional information over areas of the world not covered by local inventories in the current HTAP_v3 mosaic (e.g. Argentina (Puliafito et al. 2021), Africa (Keita et al., 2021) or the MEIC inventory (http://meicmodel.org.cn/?page_id=1772&lang=en)). Future updates of this work may also integrate reliable and up to data information over South America or Africa as time and resources permit.'*

*Similarly to the arguments reported above in the selection of the inventories contributing to HTAP_v3, in the case of MEIC the authors were not able to obtain detailed data (e.g., sector disaggregation, temporal coverage, and high spatial resolution maps) and all of the support from the MEIC experts needed to appropriately integrate the MEIC data for China into the HTAP_v3 mosaic. MEIC publicly available data (http://meicmodel.org.cn/?page_id=1772&lang=en) unfortunately do not follow the requirements of the HTAP_v3 mosaic, so enhancing cooperation with the MEIC developers is of the utmost importance in future when representing Chinese emissions. We hope to be able to include Chinese emissions from MEIC in a future HTAP mosaic inventory through the necessary cooperation with the MEIC experts.*

*In line with the comments of the Reviewer regarding specific local inventories which could have been incorporated in HTAP_v3, the following paragraph has been included in section 2.1 to clarify the choice of the selected local/regional inventories and possible future updates of HTAP_v3:*

*'Recent literature studies (Puliafito et al., 2021; Huneeus et al., 2020; Alamos et al., 2022; Keita et al., 2021; MEIC for China (http://meicmodel.org.cn/?page_id=1772&lang=en)) document additional regional/local inventories which may contribute to future updates of HTAP_v3, in particular extending the mosaic compilation to regions in the Southern Hemisphere. Considering relative hemispheric emission levels as well as the atmospheric dynamics happening in the Northern Hemisphere and regulating the transboundary transport of air pollution, the current HTAP_v3 mosaic should still satisfy the needs of the atmospheric modelling community, although improvements using latest available inventories for Africa and South America may also be considered for future updates.'*

P10 l41 – the emphasis on energy (+98%) is misleading. The total emission for this sector is not even visible in fig3a. So, here it is necessary to discuss % change in combination with absolute importance.

We removed the mention to energy since it was misleading.

Fig 3a: BC and OC will be part of PM2.5. However, they make up less than 50% of PM2.5 – some discussion on what the other 50+ % is made of? (sulfates, mineral)?

We appreciate the Reviewer's comment and we add the following explanation to the manuscript in order to clarify the difference between PM2.5 and the sum of BC and OC:

'Figure 3a shows more than 50% difference at the global level between $PM_{2.5}$ emissions and the sum of its carbonaceous components (BC and OC), which however varies depending on the region and sector. The largest difference between $PM_{2.5}$ and the sum of BC and OC is generally found for the energy and industrial sectors, where due to the high temperatures BC and OC are largely burned. Within this sector, the non carbonaceous fraction of $PM_{2.5}$ represents around 75% in Europe, 78% in the USA and up to more than 95% over Asian countries (e.g. China and India). This PM fraction is represented by other minerals, ash (mostly when burning coal) and sulphate. Road transport is also a sector showing large differences between $PM_{2.5}$ and the sum of BC and OC, with around 40% difference for Europe, around 90% difference for USA and lower values for India and China (around 15%). This component may be associated with other minerals. For the residential sector, this difference is generally lower and around 25% (for Europe and Asian countries), while around 37% in the USA and is possibly associated with other minerals and ash due to coal combustion. Shipping is also a sector where a large component of $PM_{2.5}$ (around 70%) is not associated with carbonaceous fractions but to sulphate. In particular, regions within the Sulphur Emission Control Area (SECA) show lower contributions from sulphates (e.g. Europe and USA) with an overall contribution of 5-10%. Another source of uncertainty which may contribute to enhancing the difference between $PM_{2.5}$ and the sum of BC and OC is associated on how different inventories consider condensable particulate matter.'

Also on Fig 3 panel b- are there no CO emissions from int shipping? – seems strange giving the BC, NOx contributions. Or is it just not visible?

The absolute amount of CO emissions from int. shipping is just much lower compared to the other sources (0.9 Mt in 2018), so the corresponding line is not visible in the figure.

In the supp. Material page 12 in the section S2 – Comparison of HTAP_v3 emission mosaic vs. regional and global inventories it says

*we compare HTAP_v3 against CEDS_v2021_04_21 (O'rourke, 2021), EDGARv5.0 (https://edgar.jrc.ec.europa.eu/dataset_ap50, (Oreggioni et al., 2022)), EDGARv6.1 (which is used in HTAP_v3 as gapfilling inventory, https://edgar.jrc.ec.europa.eu/dataset_ap61), etc…..*

However, in the figures S1-S4 (and legend) EDGARv6 is not present. Only v5 is represented (in green). This should be amended because now the question remains open what the difference is between v6 and the regional inventories used. Are the discrepancies the same as EDGARv5 or? Next to showing this in these valuable figures S1 -S4; the discrepancy (if present) with EDGARv6 and the regional inventories should be discussed because v6 is also used for gap filling and/or replacing any region w/o a consistent regional emission timeseries.

Figures S1-S5 now include EDGARv6 as correctly pointed out by the Reviewer.

The figures S1-S4 also  raise a few additional questions.

EDGARv5 seems to show a very strange trend for the region "islands" (e.g. CO emissions around 2012). I assume the HTAPv3 here is in line with EDGARv6? Obviously the total emission level for this region is very small but the pattern is so strange that a brief comment would be helpful

As shown in the updated Figures S1-S5, the strange pattern observed for the region 'islands' is not anymore present in EDGARv6.1. Furthermore, the difference in the comparison of SO2 and NOx emissions for the region 'Islands' is associated to the emissions of Maldives which are provided by the REAS inventory while the emissions of all the other countries are provided by EDGAR. A note is also included in the supplementary material as explanation.

**Minor corrections**

Throughout the paper (starting p2 l 34) HTAPv3 is called a tool. In my opinion it is an emission database, not a tool. I would favor correcting that throughout the MS. (e.g. a model is a tool, but an emission data base is a product in itself).  Later it is said that it also contains tools for the extraction of certain domains. Yes, those are tools to be used on the emission database. (Not tools to be used on the tool.)

We have changed the word 'tool' to 'database' throughout the manuscript.

P2 l 42-43  This is duplication of what was said before

Sentence removed.

P2 l 50 allow

Change implemented.

P3 l4 remove wide

Removed 'wide'.

P3 l8 baseline or start year?

Changed baseline to start year.

P5 l 38 – constant appears 2x

Removed the duplication of 'constant'.

P5 l39 submitted – what is status now?

This paper is now published:

Foley, K. M., Pouliot, G. A., Eyth, A., Aldridge, M. F., Allen, C., Appel, K. W., Bash, J. O., Beardsley, M., Beidler, J., Choi, D., Farkas, C., Gilliam, R. C., Godfrey, J., Henderson, B. H., Hogrefe, C., Koplitz, S. N., Mason, R., Mathur, R., Misenis, C., Possiel, N., Pye, H. O. T., Reynolds, L., Roark, M., Roberts, S., Schwede, D. B., Seltzer, K. M., Sonntag, D., Talgo, K., Toro, C., Vukovich, J., Xing, J., and Adams, E.: 2002–2017 anthropogenic emissions data for air quality modeling over the United States, *Data Brief*, 47, 109022, 33 pp., https://doi.org/10.1016/j.dib.2023.109022, 2023

P6 l5 2002 onwards

Change implemented.

P6 l20 years 200 and 2001

Change implemented.

P10 l5 replace can than consider with discuss

Change implemented.

P10 l50 should be (-24.3%) not (24.3%)

Correction implemented.

P11 l3 remove "it" from it is

Correction implemented.

P12 l 7-16 the more detailed CAMS-TEMPO dataset by Guevara et al (2021) is made specifically to match, and support the use of, the CAMS-REG inventory. So after l13 "Further analysis has shown that for the European domain 14 regional rather than country-specific monthly profiles are applied." It would be better to say: Therefore, for Europe new state-of-the-art profiles have been made available under the CAMS programme by Guevara et al. (2021).

Change implemented as suggested.

P13 l37 derives from the = is made by

Change implemented as suggested.

P14 l1 remove certain

Removed 'certain'.

P14 l2 pollutants = pollutant emissions. Moreover it would better to break l1-4 into 2 sentences.

Pollutants has been changed to pollutant emissions and the sentences have been rephrased.

Caption fig 3 delete 2018 as first word and add behind HTAP_v3 "for the year 2018"

Change implemented.

Table 1; 2[nd] row US EPA (and ECCC) is country inventory not inventories
Change implemented.

---

## Author Response (AR2)

**Public justification (visible to the public if the article is accepted and published)**:

We report here below the comments received and the answers of the authors in red.

Copernicus will want full version of DOI: https://doi.org/10.5281/zenodo.7516361. Works for me but you (and ESSD) want to save users a cut-and-paste step.

The DOI has been updated accordingly to the suggestion:

https://doi.org/10.5281/zenodo.7516361

No page numbers so hard to identify specific changes! The reviewed files included page numbers and with all modifications in track changes.

By my counting, page 7 (US EPA methods), line 25, "Foley et al." needs proper citation. 2022 or 2023? I think Copernicus imposes a necessary procedure for, or disallows, manuscripts "submitted"?

The citation of the paper by Foley et al. has been updated in the revised version of the manuscript as following:

Foley, K. M., Pouliot, G. A., Eyth, A., Aldridge, M. F., Allen, C., Appel, K. W., Bash, J. O., Beardsley, M., Beidler, J., Choi, D., Farkas, C., Gilliam, R. C., Godfrey, J., Henderson, B. H., Hogrefe, C., Koplitz, S. N., Mason, R., Mathur, R., Misenis, C., Possiel, N., Pye, H. O. T., Reynolds, L., Roark, M., Roberts, S., Schwede, D. B., Seltzer, K. M., Sonntag, D., Talgo, K., Toro, C., Vukovich, J., Xing, J., and Adams, E.: 2002–2017 anthropogenic emissions data for air quality modeling over the United States, Data in Brief, 47, 109022, https://doi.org/10.1016/j.dib.2023.109022, 2023.

Again, my counting, page 17, lines 24 and 38: Section numbering confusion, e.g. both sections labelled as 3.4?

We changed the second label of section 3.4 as 3.4.2.

One serious remaining issue: uncertainties? Uncertainty introduced in particulates sections but reader never finds a composite uncertainty or discussion of how such uncertainty might vary by pollutant? Even if regional emissions products fail to declare quantitative uncertainties (allowing you as compiler to add, multiply, etc.), compositing processes (e.g. gridding, sectoral combinations, temporal extrapolations, etc.) undoubtedly introduce additional uncertainties. No doubt final uncertainty estimate will involve mostly 'expert' judgement but readers need your best estimate! For example, can we really accept global $SO_2$ "decrease" of 100 to 73 over 20 years? What basis does reader have to accept 99.4 vs 100 or 72.9 vs 73? Or global $NO_x$ increase from 110 to 117? Users can only get trustworthy assessment of uncertainties from you. Or, in absence, they need to guess? Even a sentence or two about, or a short table of,

uncertainties globally and by pollutant? Huge effort but we all know outcome retains significant uncertainty! Tell us!

We acknowledge the remark on the uncertainty and we introduced a new section (3.5) in the manuscript to address this point, as reported here below.

**3.5 Qualitative assessment of the uncertainty of a global emission mosaic**

Assessing the uncertainty of a global emission mosaic is challenging since it consists of several bottom-up inventories and by definition it prevents a consistent global uncertainty calculation. Each emission inventory feeding the HTAP_v3 mosaic is characterized by its own uncertainty which is documented by the corresponding literature describing each dataset (see Table 2 and section 2.3) and which should be cited by the users of the mosaic for a quantitative assessment of regional uncertainties. However, the mosaic compilation process may also introduce additional uncertainties compared to the input datasets. In order to limit these additional uncertainties, we made the following considerations:

-for each emission inventory both the national totals and gridded data by sector were gathered. This process allows the mosaic compilers not to introduce additional uncertainty compared to the original input regional datasets. In fact, additional uncertainties may arise from the extraction of the national totals from spatially distributed data (e.g. country border issues which were one limitation of previous editions of the HTAP mosaics). Therefore, when regional trends are described by region and pollutant (see section 3), no additional source of uncertainty has to be considered from the mosaic compilation approach.

-the sector definition and mapping has been accurately developed following the IPCC categories and when no data was available for a certain combination of sector and pollutant a gapfilling procedure is applied using the EDGAR database. Also in this case no additional uncertainty should be considered compared to the input datasets.

-any additional uncertainty introduced by the temporal disaggregation can be deemed as negligible since each inventory already provided monthly resolution emission gridmaps and time series.

In this work we also provide a qualitative indication of the emission variability by HTAP sector and pollutant at the global level. Table S6 summarises the variability of global HTAP_v3 emissions by sector for the boundary years of this mosaic (2000 and 2018) compared to the global EDGARv6.1 data. EDGAR emissions are considered as the reference global emission inventory against which comparing the HTAP_v3 estimates although these two global products are not fully independent. The variability of the global emissions is calculated as the relative difference of the estimates of the two inventories, i.e. (EDGARv6.1-HTAP_v3)/HTAP_v3). Emission variabilities are also classified as low (L, L<15%), low medium (LM, 15%<LM<50%), upper medium (UM, 50%100%), based on the EMEP/EEA Guidebook (2019) information. The largest variability is found domestic shipping emissions (CO and NMVOC), energy (OC, BC), agricultural crops (PM), road transport (PM, NMVOC) and industry (NH3, NMVOC). In absence of a full uncertainty assessment the variability can be used as proxy of structural uncertainty, keeping in mind that variability could be biased towards overconfidence, thus underestimating the uncertainty. Furthermore, the

uncertainty of the spatial proxies has not been assessed and maybe subject of future activity updates.

Table S6 – Variability of global emission estimates by sector and pollutant, calculated as the relative difference between HTAP_v3 emissions and the EDGARv6.1 estimates. Variability ranges are based on the qualitative classes defined in the EMEP/EEA Guidebook 2019 as low (L), low medium (LM), upper medium (UM), high (H).

| Emission sector | Substance | (EDGARv6.1-HTAP_v3)/HTAP_v3, year 2000 | (EDGARv6.1-HTAP_v3)/HTAP_v3, year 2018 | varibility range, year 2000 | varibility range, year 2018 |
|---|---|---|---|---|---|
| HTAPv3_3_Energy | OC | 69.3% | 128.7% | UM | H |
| HTAPv3_3_Energy | BC | -1.9% | 77.8% | L | UM |
| HTAPv3_3_Energy | SO2 | -0.3% | 44.5% | L | LM |
| HTAPv3_3_Energy | NOx | 15.8% | 24.4% | LM | LM |
| HTAPv3_3_Energy | CO | 22.3% | 20.7% | LM | LM |
| HTAPv3_3_Energy | NMVOC | 34.9% | 15.5% | LM | LM |
| HTAPv3_3_Energy | PM2.5 | -16.4% | -1.2% | LM | L |
| HTAPv3_3_Energy | PM10 | -17.2% | -2.7% | LM | L |
| HTAPv3_3_Energy | NH3 | -1.9% | -39.5% | L | LM |
| HTAPv3_4.1_Industry | NMVOC | 59.3% | 96.4% | UM | UM |
| HTAPv3_4.1_Industry | SO2 | -15.8% | 85.5% | LM | UM |
| HTAPv3_4.1_Industry | OC | -24.0% | 50.3% | LM | UM |
| HTAPv3_4.1_Industry | BC | -3.7% | 47.8% | L | LM |
| HTAPv3_4.1_Industry | PM2.5 | -46.6% | 40.2% | LM | LM |
| HTAPv3_4.1_Industry | NOx | -1.6% | 21.5% | L | LM |
| HTAPv3_4.1_Industry | PM10 | -60.3% | -0.5% | UM | L |
| HTAPv3_4.1_Industry | CO | -25.8% | -2.6% | LM | L |

| | | | | | |
|---|---|---|---|---|---|
| HTAPv3_4.1_Industry | NH3 | -53.7% | -54.2% | UM | UM |
| HTAPv3_4.2_Fugitive | CO | 53.5% | 64.1% | UM | UM |
| HTAPv3_4.2_Fugitive | SO2 | 31.1% | 52.7% | LM | UM |
| HTAPv3_4.2_Fugitive | BC | 36.7% | 50.2% | LM | UM |
| HTAPv3_4.2_Fugitive | NH3 | 30.2% | 19.4% | LM | LM |
| HTAPv3_4.2_Fugitive | NMVOC | 10.7% | 13.4% | L | L |
| HTAPv3_4.2_Fugitive | NOx | 29.9% | 8.9% | LM | L |
| HTAPv3_4.2_Fugitive | PM10 | -0.6% | 0.9% | L | L |
| HTAPv3_4.2_Fugitive | PM2.5 | -29.0% | -23.0% | LM | LM |
| HTAPv3_4.2_Fugitive | OC | -65.0% | -51.1% | UM | UM |
| HTAPv3_4.3_Solvents | NMVOC | 2.2% | -25.2% | L | LM |
| HTAPv3_4.3_Solvents | PM2.5 | -69.8% | -60.2% | UM | UM |
| HTAPv3_4.3_Solvents | PM10 | -74.5% | -67.6% | UM | UM |
| HTAPv3_4.3_Solvents | NH3 | -99.8% | -99.6% | UM | UM |
| HTAPv3_5.1_Road_Transport | NH3 | 52.3% | 80.2% | UM | UM |
| HTAPv3_5.1_Road_Transport | NOx | -4.2% | -16.4% | L | LM |
| HTAPv3_5.1_Road_Transport | CO | -21.3% | -47.0% | LM | LM |
| HTAPv3_5.1_Road_Transport | OC | -36.2% | -51.1% | LM | UM |
| HTAPv3_5.1_Road_Transport | NMVOC | -11.0% | -58.1% | L | UM |
| HTAPv3_5.1_Road_Transport | BC | -48.3% | -60.5% | LM | UM |
| HTAPv3_5.1_Road_Transport | PM2.5 | -63.2% | -74.5% | UM | UM |
| HTAPv3_5.1_Road_Transport | SO2 | -53.1% | -81.2% | UM | UM |
| HTAPv3_5.1_Road_Transport | PM10 | -90.3% | -93.8% | UM | UM |

| | | | | | |
|---|---|---|---|---|---|
| HTAPv3_5.2_Brake_and_Tyre_wear | BC | 26.1% | 19.1% | LM | LM |
| HTAPv3_5.2_Brake_and_Tyre_wear | OC | -33.5% | -25.6% | LM | LM |
| HTAPv3_5.2_Brake_and_Tyre_wear | PM2.5 | -57.1% | -48.0% | UM | LM |
| HTAPv3_5.2_Brake_and_Tyre_wear | PM10 | -84.9% | -80.0% | UM | UM |
| HTAPv3_5.3_Domestic_shipping | NMVOC | 249.9% | 191.3% | H | H |
| HTAPv3_5.3_Domestic_shipping | CO | 221.2% | 188.7% | H | H |
| HTAPv3_5.3_Domestic_shipping | SO2 | -5.5% | 13.7% | L | L |
| HTAPv3_5.3_Domestic_shipping | PM2.5 | 11.4% | 13.6% | L | L |
| HTAPv3_5.3_Domestic_shipping | PM10 | 11.1% | 13.5% | L | L |
| HTAPv3_5.3_Domestic_shipping | BC | 5.2% | 11.3% | L | L |
| HTAPv3_5.3_Domestic_shipping | OC | 6.3% | 6.0% | L | L |
| HTAPv3_5.3_Domestic_shipping | NOx | -5.2% | 3.3% | L | L |
| HTAPv3_5.3_Domestic_shipping | NH3 | -41.5% | -20.9% | LM | LM |
| HTAPv3_5.4_Other_ground_transport | PM2.5 | -34.5% | 8.9% | LM | L |
| HTAPv3_5.4_Other_ground_transport | NH3 | -13.8% | -17.4% | L | LM |
| HTAPv3_5.4_Other_ground_transport | NOx | -55.5% | -33.1% | UM | LM |

| | | | | | |
|---|---|---|---|---|---|
| HTAPv3_5.4_Other_ground_transport | PM10 | -47.7% | -37.7% | LM | LM |
| HTAPv3_5.4_Other_ground_transport | OC | -71.8% | -41.7% | UM | LM |
| HTAPv3_5.4_Other_ground_transport | NMVOC | -80.8% | -64.6% | UM | UM |
| HTAPv3_5.4_Other_ground_transport | BC | -86.0% | -73.3% | UM | UM |
| HTAPv3_5.4_Other_ground_transport | CO | -82.6% | -82.3% | UM | UM |
| HTAPv3_5.4_Other_ground_transport | SO2 | -83.8% | -84.0% | UM | UM |
| HTAPv3_6_Residential | PM10 | 30.2% | 18.2% | LM | LM |
| HTAPv3_6_Residential | NH3 | 15.0% | 4.9% | LM | L |
| HTAPv3_6_Residential | SO2 | -8.0% | 3.9% | L | L |
| HTAPv3_6_Residential | PM2.5 | -7.4% | -9.5% | L | L |
| HTAPv3_6_Residential | NMVOC | -17.0% | -18.3% | LM | LM |
| HTAPv3_6_Residential | OC | -16.5% | -20.5% | LM | LM |
| HTAPv3_6_Residential | CO | -20.6% | -20.5% | LM | LM |
| HTAPv3_6_Residential | NOx | -39.0% | -28.8% | LM | LM |
| HTAPv3_6_Residential | BC | -41.6% | -40.3% | LM | LM |
| HTAPv3_7_Waste | NMVOC | 78.1% | 54.9% | UM | UM |
| HTAPv3_7_Waste | SO2 | 9.2% | 7.4% | L | L |
| HTAPv3_7_Waste | NH3 | -34.5% | -13.3% | LM | L |
| HTAPv3_7_Waste | PM10 | -60.8% | -48.6% | UM | LM |
| HTAPv3_7_Waste | NOx | -50.5% | -57.3% | UM | UM |

| HTAPv3_7_Waste | PM2.5 | -70.5% | -58.4% | UM | UM |
|---|---|---|---|---|---|
| HTAPv3_7_Waste | BC | -81.2% | -74.0% | UM | UM |
| HTAPv3_7_Waste | OC | -89.9% | -82.7% | UM | UM |
| HTAPv3_7_Waste | CO | -95.7% | -95.8% | UM | UM |
| HTAPv3_8.1_Agricultural_waste_burning | OC | 7.5% | 6.7% | L | L |
| HTAPv3_8.1_Agricultural_waste_burning | PM2.5 | 6.6% | 6.1% | L | L |
| HTAPv3_8.1_Agricultural_waste_burning | CO | 7.0% | 5.8% | L | L |
| HTAPv3_8.1_Agricultural_waste_burning | PM10 | 5.6% | 5.4% | L | L |
| HTAPv3_8.1_Agricultural_waste_burning | SO2 | 5.6% | 5.1% | L | L |
| HTAPv3_8.1_Agricultural_waste_burning | NOx | 5.4% | 4.9% | L | L |
| HTAPv3_8.1_Agricultural_waste_burning | BC | 3.8% | 4.0% | L | L |
| HTAPv3_8.1_Agricultural_waste_burning | NH3 | 1.0% | 2.7% | L | L |
| HTAPv3_8.1_Agricultural_waste_burning | NMVOC | -1.1% | 0.3% | L | L |
| HTAPv3_8.2_Agriculture_livestock | NOx | 11.5% | 10.7% | L | L |
| HTAPv3_8.2_Agriculture_livestock | NMVOC | -14.7% | -9.4% | L | L |
| HTAPv3_8.2_Agriculture_livestock | NH3 | -25.2% | -20.9% | LM | LM |
| HTAPv3_8.2_Agriculture_livestock | PM10 | -33.8% | -26.7% | LM | LM |

| | | | | | |
|---|---|---|---|---|---|
| HTAPv3_8.2_Agriculture_livestock | PM2.5 | -34.8% | -27.8% | LM | LM |
| HTAPv3_8.3_Agriculture_crops | NOx | 13.1% | 11.7% | L | L |
| HTAPv3_8.3_Agriculture_crops | NH3 | 16.6% | 8.7% | LM | L |
| HTAPv3_8.3_Agriculture_crops | NMVOC | 6.9% | 6.8% | L | L |
| HTAPv3_8.3_Agriculture_crops | PM2.5 | -82.1% | -77.8% | UM | UM |
| HTAPv3_8.3_Agriculture_crops | PM10 | -92.6% | -91.6% | UM | UM |

---

## Author Response (AR3)

**Public justification (visible to the public if the article is accepted and published)**:

I understand challenges around uncertainties; good on you for added Table S6 and for new section 3.5. But, back to my original question. If SO2 comparisons (EDGAR-HTAP) carry variability per sector ranging from 5 to 80%, and if - in most cases - those variabilities prove much larger for 2018 emissions compared to 2000 emissions, with what confidence can one conclude a temporal trend (decrease, 2000 to 2018) for SO2 of global 100 vs 73? Likewise but perhaps worse (lower confidence) for global NOx increase? Authors can decide what, if anything, they want to do, at proof stage. They have certainly not convinced this reader; other readers more familiar with pollutant emissions data my take different views.

We acknowledge the additional comment of the Reviewer and we take the opportunity to further clarify how to interpret the emission trends both rephrasing some sentences in the text and expanding the uncertainty section of the paper.

We added the following sentence to Section 3.1 just before the description of the trends by pollutant:

'In the following paragraphs we shortly present global and regional air pollutant emissions and their trends over the 2000-2018 period as provided by the HTAP_v3 data. Emissions are not presented with a confidence level since no comprehensive bottom-up uncertainty analysis has been performed in the context of the mosaic compilation, however see discussion in section 3.5.'

Moreover we further developed section 3.5 on emission uncertainty to clarify the doubts of the Editor and we believe that thanks to these changes and explanations the paper has strongly improved.

[revised manuscript text omitted]